

# Gravity-wave effects on tracer gases and stratospheric aerosol concentrations during the 2013 ChArMEx campaign

**Fabrice Chane Ming[1], Damien Vignelles[2], Fabrice Jegou[2], Gwenael Berthet[2], Jean-Batiste Renard[2], François Gheusi[3] and Yuriy Kuleshov[4,5]**

[1] {Université de la Réunion, Laboratoire de l'Atmosphère et des Cyclones, UMR 8105, UMR CNRS-Météo France-Université, La Réunion, France}

[2] {CNRS, LPC2E, UMR 7328, CNRS / Université d'Orléans, Orléans, France}

[3] {Laboratoire d'Aérologie, UMR5560, Université de Toulouse and CNRS, Toulouse, France}

[4] {Bureau of Meteorology, Australia}

[5] {School of Mathematical and Geospatial Sciences, RMIT University, Australia}

Correspondence to: F. Chane Ming (fchane@univ-reunion.fr)

**Abstract**

Coupled balloon-borne observations of Light Optical Aerosol Counter (LOAC), M10 meteorological global positioning system (GPS) sondes, ozonesondes and GPS radio occultation data, are examined to identify gravity-wave (GW) induced fluctuations on tracer gases and on the vertical distribution of stratospheric aerosol concentrations during the 2013 ChArMEx (Chemistry-Aerosol Mediterranean Experiment) campaign. Observations reveal signatures of GWs with short vertical wavelengths less than 4 km in dynamical parameters and tracer constituents which are also correlated with the presence of thin layers of strong local enhancements of aerosol concentrations in the upper troposphere and the lower stratosphere. In particular, this is evident from a case study above Ile du Levant (43.02°N,



6.46°E) on 26-29 July 2013. Observations show a strong activity of dominant mesoscale
inertia GWs with horizontal and vertical wavelengths of 370-510 km and 2-3 km respectively,
and periods of 10-13 h propagating southward at altitudes of 13-20 km and eastward above 20
km during 27-28 July which is also captured by the European Center for Medium range
Weather Forecasting (ECMWF) analyses. Ray-tracing experiments indicate the jet-front
system to be the source of observed GWs. Simulated vertical profiles of dynamical
parameters with large stratospheric vertical wind maximum oscillations $\pm$ 40 mms$^{-1}$ are
produced for the dominant mesoscale GW using the simplified linear GW theory. Parcel
advection method reveals signatures of GWs in the ozone mixing ratio and the specific
humidity. Simulated vertical wind perturbations of the dominant GW and small-scale
perturbations of aerosol concentration (aerosol size of 0.2-0.7 μm) are in phase in the lower
stratosphere. Present results support the importance of vertical wind perturbations in the GW-
aerosol relation. The observed mesoscale GW induces a strong modulation of the amplitude
of tracer gases and the stratospheric aerosol background.

## 1 Introduction

Gravity waves (GWs) affect dynamics of the middle atmosphere for momentum transport and
deposition, and mixing above the upper troposphere as well as chemistry and consequently
impact on global circulation and climate (Fritts and Alexander, 2003; Geller et al., 2012; Ern
et al., 2014). Several previous studies reveal atmospheric waves to be important sources of
stratospheric tracer variability. Analytical wave-tracer interaction models help to identify the
nature of waves and to quantify amplitudes and effects of waves in the middle atmosphere
from observations (Eckermann et al., 1998; Chane Ming et al., 2000; Xu et al., 2000). They
are usually used to quantify perturbations induced by GWs on tracer constituents. Small-scale
structures were misrepresented by past numerical models and therefore considered as
meteorological noise. Thanks to recent progress in computer technologies and improvements
in numerical modeling of small-scale structures, most numerical weather prediction models
and high-resolution mesoscale numerical models are now capable to produce characteristics
of GWs using GW parameterizations or by explicitly resolving convection (Shutts and
Vosper, 2011). Nevertheless, comparison between observations and numerical model outputs
shows that efforts are still needed to resolve such intermittent small-scale waves (Chane Ming
et al., 2014). Thus measuring constituent response using analytical wave-tracer interaction





models remains an inexpensive alternative way to derive correct constituent backgrounds and
wave related quantities.
Aerosols play an important role in meteorology, in radiative processes and in atmospheric
chemistry in the free atmosphere (Hobbs, 1993). Stratospheric aerosols are important for a
number of processes related to the chemical and radiation balance of the atmosphere and
consequently they impact on climate and its variability (Deshler, 2008). Even during
volcanically quiescent periods (Hamill et al., 1997; SPARC Report No. 4, 2006) when
background stratospheric aerosol loading is low, aerosols may have reduced global warming
effects (Solomon et al., 2011). During the process of formation and growth of atmospheric
aerosols, the aerosol dynamics and atmospheric chemistry are also coupled with large and
small-scale meteorological dynamics. In the troposphere, for example, vertical wind shear can
play a dominant role in aerosol-cloud interactions especially in regulating aerosol effects on
the strength of isolated deep convective clouds (Fan et al., 2009). In the stratosphere, Hommel
et al. (2015) describe how tropical stratospheric aerosol is affected by the quasi-biennial
oscillation. Vaughan et al. (1987) report dynamical influences on the short-term variability of
stratospheric aerosols from vertical profiles of aerosol backscatter ratio obtained by a high
vertical resolution lidar (30 m) and 20 min integration time at Aberystwyth (52.4°N, 4.1°W).
Their observations reveal wavelike structures which could be attributed to stratospheric GWs.
Tropical waves with periods shorter than a day could also explain some variability in aerosol
distribution observed in Indonesia (Matsumura et al., 2001). From a model study, Carslaw et
al. (1999) conclude that mountain-induced mesoscale temperature perturbations may be an
important source of nitric acid trihydrate particles in the Arctic. Bacmeister et al. (1999)
analyze the impact of mesoscale temperature perturbations and heating-cooling rates due to a
spectrum of stratospheric GWs on the growth and evaporation of stratospheric nitric acid
trihydrate particles. The microphysical trajectory-box model reveals significant scatter in
aerosol volume around the thermodynamic equilibrium values as a result of temperature
perturbations. Using a Lagrangian-based sectional box model, Nilsson et al. (2000) show that
the temperature amplitude of short period waves is the most important wave characteristic
controlling the mean nucleation rate and net particle number concentration of 0.5 nm-1µm
size classes. They also suggest that GWs could enhance the aerosol nucleation rate up to 5
orders of magnitude if the temperature perturbation due to the wave is as strong as 5 K. In
addition turbulent mixing and GWs are likely to promote the production of new aerosols
(Zahn et al., 2000; SPARC Report N°4, 2006). Thus, previous studies highlight the lack of



information on GW-aerosol interactions with measurements, especially to improve aerosol
modeling.
The present study analyzes GW-induced small-scale variability on high vertical resolution
measurements of tracer gases and stratospheric aerosols during the volcanically quiescent
period of the 2013 Chemistry-aerosol Mediterranean Experiment (ChArMEx) campaign.
Methodology and several complementary analyses based on observations and modeling are
described to detect and characterize GWs as well as GW sources. The method is fully
illustrated based on a case study on 27 July 2013. Effects of GWs on tracer gases such as
ozone and water vapor as well as GW-stratospheric aerosol relation are investigated.
The paper is structured as follows. A description of the ChArMEx campaign and the data are
presented in Sect. 2 and of methodologies and analyses in Sect. 3. Section 4 describes the
synoptic situation on 27 July 2013. Results on the case study on 27 July 2013 are discussed in
Sect. 5. Conclusions are drawn in Sect. 6.

## 2 Charmex campaign and data description

The ChArMEx campaign (http://charmex.lsce.ipsl.fr) is part of the international and regional
multidisciplinary initiative Mediterranean Integrated STudies at Regional And Local Scales
(MISTRALS). It aims at improving the characterization of short-lived (≤1 month) particulate
and gaseous tropospheric trace species responsible for atmospheric pollution over the
Mediterranean Basin, and evaluating its impacts on the present and future state of the
atmospheric environment. The campaign occurred during the Northern Hemisphere summer
from 10 June to 10 August 2013. In addition to ground-based observations, airborne
operations were organized during two special observation periods (SOPs). The first SOP from
12 June to 5 July was dedicated primarily to the study of interactions between aerosols and
radiation balance and the second one from 23 July to 9 August to the study of chemical
processes and more particularly to the formation of secondary aerosols.
Light Optical Aerosol Counter (LOAC) is a light and compact optical counter which has been
designed to perform measurements of aerosols from the ground to the middle stratosphere (up
to 37 km heights) carried by all types of balloons in various atmospheric conditions (Renard
et al., 2015). During balloon ascent, it provides information on the concentration and size
distribution of aerosols on 19 size classes from 0.2 μm to ~50 μm in diameter every ten




seconds (≈50-m heights), as well as the main nature of particles (carbonaceous aerosol,
mineral, droplets of water or sulfuric acid). The technique is based on the observation of the
scattered light by particles at two angles (Lurton et al., 2014). The instument can count up to
~ 3000 particles smaller than 1 μm, about 20 particles cm$^{-3}$ for particles larger than 1 μm in
dry conditions and up to 200 particles cm$^{-3}$ in fog/cloud conditions. For the LOAC integration
time of 10 s, the counting uncertainty at 1-sigma is about ±15 % for concentrations greater
than 10$^{-1}$ particles cm$^{-3}$. Renard et al. (2015a, b) provide a detailed description of the
instrument as well as information on calibration and validation of measurements with cross-
comparisons with different instruments during several campaigns.
Electrochemical concentration cell (ECC) ozonesondes developed by Komhyr (1969) are the
most commonly used worldwide for tropospheric and stratospheric ozone soundings (Smit et
al., 2007; GAW, 2011). The measurement principle is based on the electric current of a few
μA generated through the cell as consequence of oxidation of iodide ions by ozone contained
in ambient air bubbling in a potassium iodide electrochemical solution. The typical response
time to an ozone concentration step change is 20-30 s, so that the practical vertical resolution
for a balloon ascending at 5 ms$^{-1}$ is around 100-150 m, even though ozone data are provided
every second. The ozone measurement accuracy is within 5% in the stratosphere but is worse
in the troposphere, around 10%, owing to lower ambient ozone concentrations.
Nineteen LOAC sondes and fifteen ECC ozonesondes were launched using meteorological
balloons during ChArMEx in summer 2013, from Minorca Island (39.99°N, 4.25°E) in Spain
during SOP1 and from Ile du Levant (43.02°N, 6.46°E) in France during SOP2. Both LOAC
aerosol and ECC ozone sensors were integrated with Meteo Modem Company global
positioning system (GPS) M10 radiosondes (RSs). Temperature, horizontal wind speed and
direction are sampled every 1 s (≈ 5-m heights) and have resolution of 0.1 K, 0.15 ms$^{-1}$ and
0.1° respectively. The capacitor-type humidity sensor of M10 GPS sondes produces humidity
measurements with resolution and total instrumental error of 1% and +/-5% respectively.
Radio occultation measurements (GPS-RO) are obtained from   CDAAC   (COSMIC   Data
Analysis and Archive Center). About 120 vertical wet temperature profiles at altitudes from 2
to 40 km at longitudes of 20°W-10°E and latitudes of 30°-55°N are available between 26 and
29 July 2013. Measurements have high accuracy for temperature <1 K at heights between 5
km and 25 km and vertical resolution varying from about 0.5 km in the lower stratosphere
(LS) to 1.4 km at 40 km heights in the middle atmosphere (Kursinski et al., 1997). For a lower
limit of 1.4-km vertical wavelength ($\lambda_v$) in the upper troposphere and the lower stratosphere





(UTLS), the corresponding horizontal line-of-sight resolution, i.e., the lower limit of
horizontal wavelength ($\lambda_h$) of an observed GW is about 270 km (Kursinski et al. 1997;
Hindley et al., 2015). If the horizontal line-of-sight and the horizontal wave vector are not
aligned, GWs with shorter horizontal wavelength can be captured. Depending on the influence
of the geometric wave parameters and the measurement geometry on a homogeneous
spectrum of GWs in the range 100-1000 km horizontal and 1-10 km vertical wavelengths,
more than 80% of GW total variance of the spectrum can be derived (Lange and Jacobi,
2003). Previous climatological studies of GWs in the stratosphere have been derived from
GPS Radio Occultation (RO) data (Tsuda et al., 2000, Liou et al., 2006). In particular, Chane
Ming et al. (2014) used RO data to complement RS observations and modeling for the
characterization of mesoscale GWs produced by a tropical meteorological event: the tropical
cyclone *Ivan* over the south-west Indian Ocean.

## 3 Methodology and analyses

### 3.1 Pre-processing and noise-reduction

Vertical profiles of temperature, horizontal wind speed and ozone concentration are first re-
sampled with a 5-m vertical resolution applying a cubic spline interpolation. Profiles of
LOAC aerosol concentration are interpolated at every 50-m using a nearest neighbor
interpolation. Then, vertical profiles are filtered using a discrete wavelet transform
(DWT) also called multiresolution analysis. Multiresolution is well adapted for the analysis
and the decomposition of multiscale wavelike structures in geophysical signals (Kumar, 1997,
Domingues et al., 2005). Conventional filters often alter amplitudes and/or phase of the
original signal when filtering out high-frequency structures. Implementing finite impulse
response filters with properties such as perfect flatness in the passband, very fast decay at the
frequency cutoff and linear phase response is difficult and expensive. In the present study, the
wavelet decomposition consists of the fast application of an orthogonal filter bank, i.e., a set
of iterated so that the remaining signal (the approximation) becomes coarser and coarser after
each iteration. Then the signal can be perfectly reconstructed by adding all details and the
approximation of the signal at the $n^{th}$ order of the iteration (refer to equation 1 in Chane Ming
et al., 2000a). The smoothness or the number of vanishing moments of the wavelet are
important in theoretical and practical studies (Pollicot and Weiss, 2008) because it ensures
fast decay and limits the wavelet to capture high order polynomial behavior in a noisy signal.



But more wavelet values are needed to capture most of the energy of the signal when the
smoothness of the wavelet increases, i.e., the support of the wavelet is less compact (Walker,
2008). Compact support orthogonal bases with high orders of smoothness such as the eighth-
order Daubechies wavelet ensures energy conservation and a very good localization both in
space and frequency for the analysis of GWs in the lower and middle atmosphere.
Applications of the pyramid algorithm and the eighth-order Daubechies wavelet coefficients
are described in Chane Ming et al. (1999, 2000a). In this way, fine structures with $\lambda_v < 0.32$
km are efficiently removed from re-sampled profiles.
Perturbation profiles are then extracted from filtered data using a numerical Butterworth
infinite impulse response high-pass filter with a specific vertical wavelength cutoff between
5-7 km for temperature, horizontal wind, and ozone concentrations. Mean profiles result from
the difference between the original and perturbation profiles. A high-order Butterworth filter
is commonly used to simulate an ideal filter with maximally-flat and linear phase responses in
the pass-band. A fourth-order Butterworth filter provides a good compromise between the
cost of implementation and the frequency decay (-80 dB/decade). Because of the high
variability of aerosol concentration with altitude, the DWT is used to extract perturbations of
aerosol concentration with a vertical wavelength cutoff of 6.4 km.
Figure 1 shows examples of 50-m interpolated continuous vertical distributions of aerosol
concentrations with diameters of 0.2-50 μm in the troposphere and the stratosphere up to the
altitude of 35 km (Fig. 1b). Different shapes of vertical distribution are observed. In
particular, Figure 1c visualizes a stratospheric aerosol layer with high values of aerosol
concentration in which thin layers of strong local enhancements are embedded. Severe storms
with numerous lightning strikes occurred in France for cases in Figures 1a, 1c and 1d. The
examination of the whole aerosol dataset indicates a day to day variability of aerosols in the
UTLS. Other cases reveal that aerosol content was low in the UTLS during ChArMEx.
**3.2 Detection of GW structures**
Cases for which aerosol layers are visualized in the UTLS are analyzed by the continuous
wavelet transform (CWT) to detect GW signatures (Chane Ming et al., 2000a). The
continuous wavelet transform with Morlet complex-valued mother wavelet is applied to
perturbation profiles of temperature, horizontal wind speed, ozone and concentration of
aerosols to produce altitude-wavelength representations of the modulus of CWT coefficients,



also called scalograms. For vertical profiles in Figure 1, scalograms capture similar signatures
of wavelike structures with vertical wavelengths between 1 and 4 km in dynamical parameters
and tracer constituents of ozone and aerosols in the UTLS which are explained by the
presence of GWs. The case on 27 June 2013 is described in section 4.
The European Center for Medium range Weather Forecasting (ECMWF) model is an
operational numerical weather prediction model which can explicitly resolve stratospheric
GWs with horizontal and vertical wavelengths > 200 km and >2 km respectively (Shutts and
Vosper, 2011; Preusse et al., 2014; Chane Ming et al., 2014). Distributions of vertical velocity
and horizontal wind divergence produce snapshots of GW structures. A 1-D fast Fourier
transform (FFT) in latitude is applied on 6-hourly ECMWF operational analyses with 1.125°×
1.125° resolution at 50 hPa (about 21 km) in the LS to produce spectral densities of dominant
GWs as a function of latitudes and horizontal wavelengths.

**3.3 Characterization of GWs**

Conventional methods detailed in Chane Ming et al. (2010) such as the hodograph analysis,
the SPARC Gravity Wave Initiative Radiosonde Data and Stokes' parameter methods
(Eckermann, 1996; Vincent et al., 1997) are applied to RS vertical perturbation profiles to
provide characteristics of GWs (energy densities, momentum fluxes and spectral parameters).
The hodograph analysis computes the intrinsic frequency from the elliptical axis ratio which
is determined from variances of horizontal wind perturbations as a function of altitude.
Combined conventional methods (CCM) based on the SPARC and Stokes' parameter
methods ensure good estimation of GW spectral characteristics. The wavenumber, the
direction of horizontal propagation and the phase speed of horizontal propagation are derived
from the SPARC method and the intrinsic frequency is obtained from the Stokes' parameter
method.
The phase shift divided by the distance between two adjacent vertical profiles of temperature
in space and time at a given altitude provides the horizontal wavenumber projected along the
line connecting the two profiles. Ern et al. (2004) introduced this method to estimate global
absolute values of vertical flux of horizontal momentum from CRISTA data for horizontal and
vertical wavelengths >400 km and >5 km respectively. The method is adapted to pairs and
triads of RO temperature profiles using the S-transform and CWT to estimate the phase shift
of horizontal wavelengths for horizontal and vertical wavelengths >1500 km and >5 km
(Wang and Alexander, 2010; Faber et al., 2013). In the present study, RS and RO temperature
profiles are combined to analyze GWs with shorter horizontal and vertical wavelengths than
those described in previous studies.
**3.4 Simulated GWs**
The GW parameters are used to initialize a model based on the simplified linear wave
polarization relations for vertically propagating internal waves in a rotating fluid (Gill, 1982;
Chane Ming et al., 2002) to produce synthetic vertical profiles of temperature and horizontal
winds with GW signatures. Input parameters are vertical wavelength, intrinsic frequency,
direction of horizontal propagation, localization, altitude range, amplitude of temperature
perturbation and mean profiles of temperature and horizontal winds. A Gaussian-shaped
sinusoidal pulse is first computed at the observed altitude range from temperature
perturbations. Based on the simplified linear wave polarization relations, vertical wind
perturbations are derived from temperature perturbations. Then the horizontal wind
perturbations result from the vertical wind perturbations when the rotated frame of reference
is aligned along the horizontal wavenumber. The direction of horizontal wave propagation is
finally used to rotate the frame of reference to obtain observed horizontal wind perturbations.
The simulated profiles result from the sum of perturbations and observed background profiles.
**3.5 Ray-tracing**
The Gravity Wave Regional or Global Ray Tracer (GROGRAT) is a four-dimensional ray-
tracing model of the propagation and amplitude evolution of non-hydrostatic GWs in rotating,
stratified, compressible slowly-varying background described by numerically gridded fields
of temperature, wind and pressure or geopotential height in the lower and middle atmosphere.
It is based on the dispersion and refraction of the ray paths and includes parameterizations of
turbulent and radiative wave damping and of wave-amplitude saturation. A full description of
the model can be found in Marks and Eckermann (1995) and Eckermann and Marks (1997).
The ray path stops when the vertical wavelength becomes too large or varies too quickly with
altitude, in particular near GW sources. Previous studies have shown that GROGRAT is an
efficient tool to identify GW sources (Guest et al., 2000, 2002) and to simulate GW-
background interactions such as GW effects, wave filtering, space and time variability of GW
activity and characteristics. For example, GROGRAT simulations help to interpret global GW





observed morphology derived from satellite observations (Preusse et al., 2009). In the present
study, the model GROGRAT is run with observed GW spectral characteristics in a regional
background of 6-hourly operational ECMWF analyses of temperature, horizontal wind and
geopotential height with resolutions of 25 pressure levels (1000-1 hPa) and 1.125°x1.125° in
the horizontal plane.

### 3.6 Parcel advection method

The parcel advection method is based on an analytical formula for the linear response of
minor constituents to non-dissipating hydrostatic internal GW motions when photochemical
response of the constituent can be negligible, especially for long lived tracers. The method
was used for GW-induced perturbations in ozone, water vapor and other constituents such as
nitrogen dioxide and sodium (Randel, 1990; Teitelbaum et al., 1994).
When the photochemical response of the constituent is negligible, in the linear simplified
case, normalized ozone mixing ratio perturbations as a function of altitude are linked to
normalized potential temperature perturbations by a coefficient R(z) which is the ratio of
vertical gradients of ozone and of the potential temperature background. Chane Ming et al.
(2000b) illustrated the method to produce a climatology of laminae with vertical wavelengths
<5 km induced by GWs and horizontal advection in the UTLS near the southern subtropical
barrier. The method also revealed the presence of GWs on vertical ozone profiles collected at
Macquarie Island (54°S, 159°E) during the Airborne Southern Hemisphere Ozone
Experiment/Measurements for Assessing the Effects of Stratospheric Aircraft
(ASHOE/MAESA) observation programme in 1994 (Chane Ming et al., 2003). Eckermann et
al. (1998) formalized the method to derive analytical formulas for the response of vertical
constituent profiles of arbitrary shape to adiabatic GWs. In particular, they showed that the
parcel method becomes inaccurate for non-hydrostatic GWs and if the wave-induced
photochemical response produces a significant diabatic feedback on the GW for shorter-lived
constituents.

### 4 Synoptic situation on 27 July 2013

A heat wave accompanied by local severe thunderstorms affected the whole of France during
July 2013, especially during the second part of the month from 15 to 28 July with local
maximum temperatures > 35°C on 25-27 July. Météo-France ranked it the third hottest July



since 1900. As for thunderstorm activity, a total of 527,496 lightning strikes were registered.
The most severe stormy episode of the year occurred on 27 July 2013 when both synoptic
observations and forecasts indicated the presence of a meteorological pattern similar to an
upper level jet streak. A jet streak is characterized by localized wind maxima along the axis of
a jet stream and often referred as a jet-front system because it is generally found in association
with frontal zones in the UT (Cunningham and Keyser, 2003). Right entrance and left exit
regions of jet streaks (with right and left set by the axis of the jet core and defined facing
downstream) are preferred areas of storm development characterized by front systems (Lin,
2007). Thus, such feature plays a role in the development of mesoscale convective complexes.
Climatologies at latitudes of North America reveal that jet streaks are intrinsically linked to
tornado and severe storm reports (Rose et al., 2004; Clarke et al., 2009). From a
climatological survey of such synoptic patterns, Uccellini and Koch (1987) show the
existence of mesoscale GW disturbances which can severely impact on the weather
conditions. For operational forecasting and detection of such GWs, Koch and O'Handley
(1997) propose a conceptual model with location of GW occurrence in a diffluent region
bounded by the 300-hPa ridge axis to the northeast and the 300-hPa inflection axis to the
south-west. The southern boundary of the wave region is defined by the location of a surface
warm front. Numerical mesoscale studies also support exit regions of jet-front systems at the
cold side of surface front as a favorable area of GW generation (O'Sullivan and Dunkerton,
1995; Zhang, 2004). Recently Plougonven and Zhang (2014) presented a review of current
knowledge and understanding of GWs near jets and fronts from observations, theory, and
modeling including jet streaks.
On 27 July 2013, mean wind speed was about 150 kmh$^{-1}$ near the tropopause above the
western part of France. A sequence of 15-min georeferenced infrared GMS-3 satellite images
reveals increasing convective activity associated with the jet-front system on 27 July from
0300 UTC. Very warm, moist and unstable air coming from the south triggered a strong
mesoscale storm like a bow structure along the north-west coast of France propagating north-
eastward in the afternoon of 27 July.
Figure 2a depicts the meteorological situation on 27 July at 1200 UTC derived from
operational ECMWF analyses, which represents a typical jet streak (refer to Figure 2 in Koch
and Hadley (1997)). Isohypses on a 500-hPa meteorological chart visualize the synoptic
pattern with a strong minimum of geopotential heights located at 45°N of latitude and 15°W
of longitude which strengthens the jet wind field. Maximum wind speed (>50 ms$^{-1}$ from 1200



UTC) at 300 hPa located west of Portugal indicates the presence of a significant jet streak.
Height-longitude cross-section of potential vorticity at 43°N (not shown) shows the presence
of a tropopause fold (6°W) located west of the jet (4°W). This observation is supported by the
cross-section of ozone mass mixing ratio visualizing a stratospheric intrusion of ozone in the
middle troposphere at longitudes between 6°W and 2°W. Turbulent mixing processes within
the tropopause fold are first-order importance as a mechanism for stratospheric-tropospheric
exchange (Shapiro, 1980). The jet streak dissipated on the afternoon of 28 July.
Backward air mass trajectories are calculated with the FLEXible TRAjectories (FLEXTRA)
5.0 code, a kinetic trajectory calculation code (Stohl et al., 1995, 2005). Trajectories were
initialized with global 3-hourly ECMWF wind fields with 1°x1° horizontal resolution during
10 days from 27 July at 2300 UTC at every 1-km height between 15 km to 25 km above Ile
du Levant (Fig. 2b). Trajectories reveal two different eastern air masses in the LS originated
from tropical UT and mid-latitude LS. The vertical profile of aerosol concentration at Ile du
Levant shows evidence of two layers of large amount of aerosols with diameters between 0.2
and 50 μm at altitudes of 15-20 and 20-25 km heights and a minimum at 20 km heights where
the zonal wind reverses (Fig. 3). During volcanically quiescent period, previous studies
mention the presence of a stratospheric sulfate aerosol layer also called the Junge layer with a
quasi-steady state relative maximum in particle number concentration (r> 0.1μm) around 20
km in the mid-latitudes (Junge et al., 1961). Recent observations also reveal the presence of
carbonaceous material in stratospheric aerosol (Murphy et al., 2007). In the present study, the
"speciation index" retrieved from the 12 and 60° channels LOAC measurements suggests that
the nature of stratospheric aerosols with diameters <1 μm is mainly carbonaceous in the LS
on 27 July above Ile du Levant (Renard et al., 2015a). Low aerosol content is observed in the
LOAC profile of aerosol concentration on 28 July at 1524 UTC when the jet streak dissipates
after 1200 UTC. Thus, dynamical processes might be involved in shaping the local
stratospheric aerosol background.
**5 Results**
The methodology and the analyses described in section 3 are now illustrated on the case study
of 27 July 2013.
**5.1 Detection of GWs**



Figure 3 visualizes filtered resampled vertical profiles of temperature, potential temperature,
horizontal winds and ozone from radiosondes and the vertical profile of LOAC aerosol
concentrations with diameters of 0.2-0.7 μm on 27 July at 2303 UTC above Ile du Levant.
In our study, GW analyses are applied on the sum over the size classes between 0.2 and 0.7
μm of continuous vertical profiles of aerosol concentrations to derive a mean continuous
vertical profile of aerosol concentrations. Because of the synoptic meteorological event, a
sheared tropospheric jet is observed at 10-km heights with maximum wind velocities <20 ms$^{-1}$
at altitudes of 6 km and 14 km while larger amplitudes (≈40 ms$^{-1}$) are observed at longitudes
<2.5°W at heights of about 12.5 km. Perturbation profiles are obtained from subtraction
between original and background profiles (Chane Ming et al., 2010). They show evidence of
wavelike structures with short vertical wavelengths of 2-3 km in temperature and horizontal
wind at heights of 13-20 km. Maximum temperature and wind perturbations are about 3°C
and 5-7.6 ms$^{-1}$ respectively near the tropopause located at the height of 15-km. Large variation
in ozone and aerosol perturbations are also observed at altitudes of 15-20 km. Thus the CWT
is applied on vertical profiles of perturbations to localize wavelike structures in altitude-
wavelength representations at heights below 23 km (refer to section 3.2). The scalograms of
the dynamical variables (temperature, wind) reveal a spectral signature with 2.6-km vertical
wavelength at heights 10-20 km (Fig. 4a, b, c). For ozone and aerosols, similar peaks are
found above 15 km (Fig. 4d, e). Such pattern is also observed on CWTs of each vertical
aerosol concentration profile of the five smallest size classes. Analyzing normalized ozone
perturbation profiles enhances ozone peak at 13 km heights because of low level tropospheric
ozone background whereas the use of ozone perturbation profiles focuses on wavelike
structures observed at 15-23 km heights. The cross-sections of potential vorticity and ozone
mixing ratio at 43°N indicate that the peak at 13 km heights results from a local stratospheric
intrusion at longitudes of 7-12°W. The longitude-latitude maps at 250-300 hPa support the
presence of a filament structure of potential vorticity and ozone mass mixing ratio during the
jet-streak event (not shown). These observations also suggest that aerosols observed on the
LOAC profile at the same altitude are of stratospheric origin on 27 July at 1200 UTC (Plumb
et al., 1994).
The Morlet CWT local maxima also called "skeleton" (Chane Ming et al., 2000a) provide a
continuous distribution of the dominant wavelike structure with the vertical wavelength of 2-3
km at heights between 3 km and 25 km with maxima at heights of 13-18 km in zonal wind
perturbations (not shown). The vertical wavelength is estimated at about 2.5 km at altitudes of





3-7 km. The amplitudes as well as the vertical wavelength decrease rapidly
at altitudes above 20 km likely because waves are both filtered at the critical level where
intensity of zonal wind approaches zero. The FFT also supports the presence of a dominant
wavelike structure with vertical wavelengths of 2-3.5 km at heights of 13-23 km in all
analyzed parameters (Fig.4f).
The DWT decomposition reveals that RS horizontal wind components (u and v) with
dominant vertical wavelengths of 2-3 km are in quadrature at heights of 14-26 km in
agreement with the linear GW polarization relation. The hodograph of wind perturbations
visualizes elliptical structures with characteristics of GWs with vertical wavelength of about 2
km at heights of 14-24 km. The sign change at heights of 9-12 km with anticyclonic
(cyclonic) rotation of the ellipse above 12 km (below 9 km) indicates an upward (downward)
flux of GW energy. This suggests that GW sources are localized at heights between 9 and 12
km.
The analyses of vertical velocities in pressure coordinates ($\omega = dp/dt$) derived from ECMWF
operational analyses indicate that the numerical weather prediction model captures wavelike
structures on 27 July 2013 at 1800 UTC in the LS (Fig. 5). Figure 5a depicts a mesoscale
structure located over the eastern part of France. Spectral density of vertical velocities is
calculated for longitudes of 10°W-20°E at every latitude step. In particular, the energy of
mesoscale wavelike structures with horizontal wavelength in the east-west direction $\lambda_x < 500$
km is well-localized at the latitude of 46 °N which corresponds to the latitude of the exit
region of the jet streak between latitudes of 42° and 52°N (Fig. 5b). The figure also suggests
the presence of longer horizontal structures (500 km$<\lambda_x<$2000 km) at latitudes of 45-52°N.
But estimated horizontal wavelengths might be biased at latitudes outside the interval 42° and
47° where the observed mesoscale structures could not be considered
to be plane waves with north-south oriented wave fronts. Figure 5c displays the energy
distribution of dominant wavelike structures at latitudes of 42-52°N for horizontal
wavelengths $\lambda_x$ <1500 km. In particular, it highlights modes with horizontal wavelengths $\lambda_x$
of 400-800 km with a dominant mode of 400 km at latitudes of about 46°N. The spectral
densities at latitudes of 42-48°N and 46°N confirm that the mode of 400 km is dominant at
46°N within the wavelength range of 400-800 km (Fig. 5d). The vertical velocity data also
reveals that the structure with amplitudes < 0.2 Pas$^{-1}$ extends at longitudes (latitudes) between
4°W to 10°E (42°N to 52°N).




The RO data from 26 July to 29 July 2013 are examined to study the energy activity of
wavelike structures with vertical wavelengths of 1.5-5 km (Fig. 6a). The RO soundings over
France are mostly located at longitudes of 2.5°-6°E and latitudes of 40-50°N where wavelike
structures are well observed in the ECMWF analyses. The spectral analyses of RO
temperature profiles support the presence of a strong energy activity of wavelike structures
($\lambda_v$<5 km) in the area. Figure 6b indicates that spectral density peaks for dominant modes
with vertical wavelengths of 2-3.5 km, especially from 27 July at 0600 UTC to 29 July in the
morning with a maximum observed on 28 July early in the morning.
Thus both observations and ECMWF analyses suggest that the strong activity of GWs with
vertical and horizontal wavelengths of 2-5 km and <1700 km with a dominant mesoscale
structure of 400-km horizontal wavelength is associated with the jet-front system from 27 July
afternoon to 28 July in the morning in the eastern part of France (refer to the following section

13   5.2).

## 5.2 Spectral characteristics of GWs

The hodograph analysis and the CCM (refer to section 3.3) are applied on RS temperature and
wind perturbations on 27 July at 2303 UTC above Ile du Levant to compute spectral
characteristics such as vertical and horizontal wavelengths, intrinsic frequencies, periods,
directions of horizontal wave propagation and wave energy densities (Table 1). Consistent
spectral parameters are obtained, in particular the two methods detect a mesoscale inertia GW
with vertical and horizontal wavelengths of 2.6 km and 440-510 km respectively, a period of
about 12 h, propagating southward at heights of 13-20 km. Eastward dominant modes with
similar vertical wavelength, a period of about 15 h and longer horizontal wavelengths ($\approx$1300
km) are observed at heights of 20-26 km. The horizontal phase speed of the mesoscale GW is
estimated at about 6.8 ms$^{-1}$ (7.9 ms$^{-1}$) at heights of 13-20 km (20-26 km).  GWs with 1300-km
horizontal wavelength have larger phase speed of about 24 ms$^{-1}$ at heights of 20-26 km.
Consistent large values of total energy densities of about 16 J.kg$^{-1}$ and vertical flux of
horizontal momentum of 0.05 m$^2$s$^{-2}$ (vertical flux of zonal momentum $\overline{u'w'}$ $\approx$0.045 m$^{-2}$s$^{-2}$)
defined in Vincent et al., (1997) are calculated at heights of 13-20 km. In particular the kinetic
energy densities are about three times as large as potential energy densities (spectral
index$\approx$2.6-2.9) which means an excess of wave energy near the inertial frequency (Vincent et
al., 1997; Hertzog et al., 2002). A large decrease of total energy densities above heights of 20





km indicates that most tropospheric GWs are not transmitted to higher stratospheric layers.
The wave energy propagates upward from the troposphere with a fraction of upward energy
Fup>86 % (95%) at heights of 13-20 km (20-26 km). In addition the hodograph analysis
reveals the presence of a mesoscale structure propagating south-westward with vertical and
horizontal wavelengths of 2.6 km and 161-451 km respectively with periods of 10-15h at
heights of 3-7 km in the lower troposphere. The Fup value of about 60% suggests that wave
energy equally propagates upward and downward below the dominant GW tropospheric
source. Finally, conventional methods provide good estimates of spectral GW parameters
as can be seen from consistent values of the two methods and the standard deviation
computed for the hodograph analysis. Similar results are also obtained by the CWT method
described in Chane Ming et al. (2002, 2003).
The horizontal wavelength is estimated from a triad of temperature profiles from RS at 2303
UTC (hereafter called RS2723) on 27 July above Ile du Levant and RO at 0200 UTC
(hereafter called RO2802) and 1200 UTC (hereafter called RO2812) on 28 July(Fig.7a). Here,
a high-resolution RS profile is used as a reference for identifying wavelike structures with
vertical wavelengths of 2-4 km in RO data. The vertical profiles of temperature perturbation
visualize GW signatures at heights of 10-20 km. Phase shifts ($\Phi$) between perturbation
profiles are directly calculated for each pair of profiles at heights of 15-20 km where a
dominant wavelike structure is detected in the RS profiles (refer to Table 1). Horizontal
wavelengths are estimated at about 430 km ($\Phi$=2.65 radians, distance=179.14 km), 551 km
($\Phi$=5.24 radians, distance=459.53 km) and 560 km ($\Phi$=3.14 radians, distance=280.4 km)
from the three pairs of vertical profiles respectively (RS2723 and RO2802, RS2723 and
RO2812, RO2802 and RO2812). The minimum value of 430 km is the nearest estimation of
the real value of the horizontal wavelength (Faber at al., 2013). The phase shift is also
computed using the CWT of temperature perturbations. Both figures 7b and 7c show evidence
of a similar dominant mode at heights of 10-18 km with vertical wavelengths of 2.5-3 km as
the cross-wavelet spectrum depicted on Figure 7d. Phase shift is derived from the cross-
wavelet spectrum for the observed dominant mode with vertical wavelengths of 2.5-3.5 km
and converted to horizontal wavelength. The second panel of Figure 7d shows evolution of
the estimated horizontal wavelength as a function of altitude for vertical wavelengths of 2.5-3
km. The horizontal wavelengths vary between 400 km and 680 km at heights of 10-20 km. An
additional phase error may be introduced between GPS RO and the radiosonde/GPS RO pairs
due to radiative transfer and retrieval (Preusse et al., 2002). Using Eq. 7 reported in Ern et al.



(2004), the vertical flux of momentum is estimated at 0.03 $m^2s^{-2}$ at heights of 10-20 km. It
underestimates the vertical flux of momentum of 0.05 $m^2s^{-2}$ previously computed for this
case. Characteristics of GWs are consistent with previous observations of inertia GWs
associated with a jet stream exit region (Thomas et al., 1999; Ravetta et al., 1999; Bertin et al.,
2001). The observed characteristics are used in the next section to provide a linear model of
the mesoscale GW and to identify its source.

**5.3 Simulated profiles and ray-tracing**

The spectral parameters (a vertical wavelength of 2.6 km, a period of 11.7 h, a direction of
horizontal wave propagation clockwise from north of 204°) at heights of 12.8 km with a
height range of 14 km are used to produce synthetic RS profiles with the signature of such a
dominant mesoscale GW with upward propagation of energy (refer to section 3.4). Because of
the variation of the Brunt-Väisälä frequency with height, the horizontal wavelength, computed
using the GW dispersion relation, has values of 225.3 ±22.6 km and 499.9 ±45.7 km at
heights of 3-7 km and 13-20 km respectively. A value of 392.7 ±13.6 km is estimated at
heights of 13 km. Filtered and simulated amplitudes, and phase relations of perturbations are
in good agreement (Fig.8). The hodograph analysis and the CCM are applied on simulated
profiles to retrieve spectral parameters (Table 1). The two methods provide good estimates of
the original spectral parameters as well as energy densities at heights of 13-20 km. The wave
energy propagates upward from the troposphere with a fraction of upward energy Fup>93 %
and a horizontal phase speed of 6.44 $ms^{-1}$ at heights of 13-20 km. In particular the structure
has a horizontal wavelength of about 200 km in the lower troposphere. In conclusion, results
indicate that the mesoscale inertia GW structure agrees well with simplified linear GW theory
and that it is the most dominant structure observed in the RS profiles on 27 July. Because of
the shape of the wavelike structure, the Morlet complex-valued mother wavelet reveals to be
well-adapted to analyze such a structure.
Taking into account uncertainties of spectral parameters (Table 1) obtained by the two
methods, spectral characteristics of the mesoscale inertia GW with horizontal wavelengths of
300-550 km (step of 20 km), ω/f of 1.4-1.8 (step of 0.1) and horizontal propagation direction
of 200° at heights of 19 km from 28 July at 0000 UTC are used to define a discrete spectrum
of 65 individual components to initialize the GROGRAT model (refer to section 3.5).
Backward ray trajectories are computed with the same parameterizations used in Guest et al.





(2000) and a time step of 6 min. Almost all rays can be traced back until 12 h-17 h before
release on 28 July at 0000 UTC. Projected ray paths on the georeferenced infrared GMS-3
image at 1200 UTC and the latitude-height cross section of zonal wind at longitude 0° at 1200
UTC reveal that ray paths terminate at the location of the cold frontal zone and at about
heights of 10 km in the jet core on 27 July at 1200 UTC at longitudes of 2°-0°W and latitudes
of 42.5°-45°N (Fig. 9a, 9b). Figure 9c shows the evolution of the mean horizontal wavelength
and the normalized intrinsic frequency as a function of altitude and mean time. The horizontal
wavelength (period) decreases (increases) slowly from 440 km (10 h) at 17 km heights to 290
km (14 h) at 10 km heights. The time of vertical progression of GWs is twice as long at
altitudes of 16-19 km which ensures GW propagation far away on a horizontal plane in the
LS. A sensitivity test on the horizontal propagation direction varying between 170-230° (with
a step of 10°) produces similar results. In conclusion, previous results derived from both
observations and modeling support that the mesoscale inertia GW at heights of 13-20 km is
generated by the jet-front system. They are consistent with the studies of Zhang (2004) and
Wang et al. (2009). Investigating mechanisms of generation of GWs in upper tropospheric jet-
front systems using a MM5 mesoscale model, these studies reveal the production of
mesoscale GWs lasting more than 24 h with very similar spectral characteristics ($\lambda_h$=100-500
km, $\lambda_v$=2 km, $\omega$/f≈1.5) in the troposphere and LS.
**5.4 GW induced perturbation in tracers and aerosol concentration**
The parcel advection method (refer to section 3.6) is applied on profiles of normalized ozone
and potential temperature perturbations on 27 July at 2303 UTC (Fig. 10a). The normalized
potential temperature perturbations are multiplied by smooth vertical profiles of coefficient
R(z) (Fig. 10b) and superimposed on normalized ozone perturbations. The amplitudes of
perturbations exhibit comparable values at heights of 4-7 km and 14-24 km for perturbations
with vertical wavelengths <5 km. This supports the presence of GW signatures. The parcel
advection method is not valid at heights of 7-14 km because the vertical gradient of the ozone
background is very small, R(z) is strongly variable with altitude, and other tropospheric
structures are present at 7.5 km and 12.5 km (Chane Ming et al., 2000b). The parcel method
also identifies GW signatures in temperature and ozone perturbation profiles on 28 July at
1330 UTC (Fig. 10c and 10d). In comparison with Figure 3, smaller values of normalized
potential temperature perturbations are observed in the LS. It means that the intensity of the
stratospheric GW is decreasing in the afternoon of 28 July. The Morlet CWTs of ozone and



temperature perturbations depict similar monochromatic structures with vertical wavelength
of about 1.5-3 km at heights of 10-15 km. For these two cases, GW signatures can be
observed in the lower troposphere above 3 km. The parcel advection method is also valid for
temperature and humidity mixing ratio. Figure 10e visualizes wavelike structures on specific
humidity profiles at heights above 3 km. The spectral densities of perturbations (Fig. 10f)
reveal wavelike structures of 1.5-3 km vertical wavelengths. The amplitudes of temperature
and specific humidity perturbations are $\pi$ out of phase above 4-km height which is in
agreement with the parcel advection method with regards to signs of mean gradients. Thus,
evidence of GW signatures in ozone and humidity in the lower troposphere supports that the
dominant GW source is located in the jet core with an upward (downward) propagating
energy above (below) the source. Thus observations lead to the conclusion that the mesoscale
GW observed on 27 July 2013 affected the variability of stratospheric ozone-tracer fields.
The DWT is applied to aerosol concentrations to retrieve perturbations of aerosol
concentrations with vertical wavelengths of 1.6-3.2 km and the mean profile with vertical
wavelengths >3.2 km (Fig.11). A dominant wavelike structure with vertical wavelength of 2-
2.5 km is observed in the LS (15-20 km). The amplitudes of aerosol concentration
perturbation strongly decrease at heights of 20 km likely because of partial transmission of
GWs through the wind field and the presence of two layers of stratospheric aerosols with
different origins. The superimposition of simulated vertical perturbation aerosol upon
concentration perturbations profiles of the dominant mesoscale GW (Fig. 8b) indicates that
perturbations of aerosol concentration and simulated vertical wind are in phase at heights of
15-19 km (Fig. 11b). The phase relationships do not agree with the parcel advection method
for which perturbations should be in quadrature (refer to equation 24 in Eckermann et al.,
1998). Aerosols behave differently from tracer gases with production and growth processes in
the stratosphere. This case study suggests a possible correlation between vertical transport and
the distribution of aerosol concentration in the LS. In addition, large amplitudes of
stratospheric vertical wind with maximum values of $\pm$ 40 mms$^{-1}$ are induced by the mesoscale
GW in comparison with monthly averaged amplitudes of stratospheric vertical wind of about
$\pm5$ mms$^{-1}$ with an annual averaged of $\pm1$ mms$^{-1}$ (Gryazin and Beresnev, 2011). Recent
theoretical studies reveal possible substantial effects of a background vertical motion on
stratospheric aerosol distribution. Using a four-dimensional continuity equation for particles
undergoing growth process, Li and Boer (2000) investigate the relative roles of condensation,
particle fall velocity, vertical motion, and diffusion in determining the aerosol size




distribution. Vertical motion could broaden the size distribution of the stratospheric
background aerosol which would tend to support a broader size distribution of the
stratospheric background aerosol in the tropics. Gryazin and Beresnev (2011) suggest that the
relative aerosol concentration is mostly controlled by the action of the vertical wind for
particles with a size < 2.5 μm. Nilsson et al. (2000) propose the use of the phase shift in
perturbations of aerosol concentration and vertical wind to separate the influence of turbulent
fluxes of aerosols from the effect of nucleation responding on waves or turbulence.
Perturbations should be $3\pi/4$ out of phase if nucleation dominates whereas turbulence would
cause perturbations to be in phase. With regard to GWs effects on cirrus clouds, the
combination of a mountain wave and the jet-streak GW producing a wave-induced upward
motion > 10 cms$^{-1}$ can cause ice supersaturation and trigger the formation of a cirrus cloud in
the UT (Spichtinger et al., 2005). A strong link has been established between GW-induced
mesoscale variability in vertical velocities and climate forcing by cirrus (IPCC, 2007). Haag
and Kärcher (2004) investigate the impact of aerosols and GWs on cirrus clouds at Northern
mid-latitudes. They emphasize the importance to include small-scale temperature fluctuations
caused by GWs for a good prediction of cloud physical properties in global models as well as
to represent correctly changes in GW activity in a future climate. In the present study, strong
modulation is also observed on amplitudes of stratospheric aerosol concentration background
with size class of 0.2-0.7 μm up to 60% of the background amplitude with the vertical spectral
characteristic of the observed stratospheric mesoscale GW.

## 6 Conclusions

Model studies reveal that GWs are likely to play a direct or an indirect impact on aerosol
processes. In addition the influence of GWs should be important during volcanically
quiescent periods because of low aerosol loading, especially in the tropics where the spectrum
of GWs and production of stratospheric aerosols are larger. A good knowledge of GW sources
is also crucial for determining GW spectral characteristics and intensity which will have an
impact on stratospheric aerosols. However, observations of effects of GWs on stratospheric
aerosols are still poorly documented mainly because of aerosol instrumentation limitations.
These observations are needed to improve aerosol modeling and to understand effects in
climate models. Balloon-borne optical particle counters provide direct *in situ* measurements
of stratospheric aerosol concentrations in several size classes. They have been shown to be
useful for aerosol observations during quiescent periods because of the low aerosol loading





and the presence of small size aerosols (Rosen et al., 1975). Performance of such instruments
with vertical resolution of about 5-m height has been recently improved to understand short-
scale variability in aerosol distribution.
High-resolution observations of the new balloon borne LOAC coupled with RS and
ozonesonde observations are shown to capture short-scale wavelike structures in the UT and
LS during a volcanically quiescent period during the 2013 ChArMEx. The present study
proposes a methodology and different complementary tools based on observations and
modeling, to describe GWs and to evaluate their effects on tracer constituents and the vertical
distribution of aerosol concentration. The methodology is fully illustrated on a case study on
27 July 2013 when a mesoscale inertia GW produced by the jet-front system was identified
during a jet-streak event. Such GWs induced strong local short-scale perturbations in the
amplitudes of tracer constituents, such as ozone and water vapor as well as the stratospheric
aerosol background layer. The simulated profile of stratospheric vertical velocity
perturbations produces a wave-induced upward motion with a maximum amplitude of 40
$mms^{-1}$ more than ten times larger than usual amplitudes of stratospheric vertical wind. The
profile of wave-induced upward motion reveals to be in phase with perturbations of
stratospheric aerosol concentrations (size class of 0.2-0.7 µm). Besides the role of temperature
perturbations especially for nucleation at high frequencies, present results also support the
importance of GW spectral characteristics and GW-induced vertical wind perturbations in the
vertical aerosol distribution as also observed in formation of ice supersaturation below the
tropopause where cirrus clouds are observed (Spichtinger et al., 2005). In addition our study
lead to the conclusion that mesoscale GWs might affect significantly the mean vertical
distribution of stratospheric aerosol concentrations.
A strategy of highly-regular balloon launches has been set up using the LOAC instrument in
particular to investigate the short-term variations of the vertical profile of the tropospheric and
stratospheric aerosols using weather balloons in the framework of VOLTAIRE (VOLatils-
Terre, Atmosphère et Interactions-Ressources et Environnement) since December 2013.
Balloon-borne LOACs are launched regularly (twice per month) at Northern mid-latitudes and
occasionally at other latitudes. In our future research, this analysis will be extended to a
substantial dataset to investigate the occurrence of GW effects on LOAC vertical profiles of
stratospheric aerosol concentrations. These preliminary results do not exclude possible
threshold on GW amplitudes to trigger the GW-aerosol relation. In future, mesoscale
modeling with the French Meso-NH model coupled with an analytical stratospheric aerosol



model will enable us to highlight the different processes and the efficiency of such adiabatic
reversible transient events to describe the stratospheric aerosol background in particular when
GWs vanish.
**Acknowledgements**
The LOAC project was funded by the French National Research Agency's ANR ECOTECH.
The LOAC and the gondola were built by the Environnement-SA and Meteo Modem
companies. The balloon flights of the ChArMEx campaign were funded and performed by the
French Space Agency CNES. The ozone and LOAC sondes used in the campaign were
funded with the support of CNES, ADEME, and INSU-CNRS in the framework of the
MISTRALS Programme. This scientific work was financially supported by the
French Labex VOLTAIRE (Laboratoire d'Excellence ANR-10-LABX-100-01) and the French
project, StraDyVariUS ANR-13-BS06-0011-01. The satellite images were provided by
Météo-France/Centre de Météorologie Spatiale. Dr Stephen D. Eckermann provided
the GROGRAT software. The model was run on the supercomputer of the University of La
Réunion.

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





Table 1. Characteristics of GWs on 27 July at 2303 UTC above Ile du Levant using a hodograph analysis and combined conventional methods (grey cells) on observed and simulated (marked by *) profiles. Height: altitude range, $E_t$: total energy density, $E_k$: kinetic energy, $E_p$: potential energy, Phi: direction of horizontal wave propagation (clockwise from north), $\lambda_v$, $\lambda_h$: vertical and horizontal wavelengths, $\omega/f$: intrinsic frequency/inertial frequency, T: period. Parentheses denote standard deviation. Units are in brackets.

| Heights (km) | $E_t$ (J.kg$^{-1}$) | $E_k$ (J.kg$^{-1}$) | $E_p$ (J.kg$^{-1}$) | $\lambda_v$ (km) | $\lambda_h$ (km) | $\omega/f$ | T (h) | Phi (°) |
|---|---|---|---|---|---|---|---|---|
| 3-7 | 11.3 | 7.6 | 3.7 | 2.6 | 451 (23) 204 (10) 161 (20) | 1.2 (0.1) 1.6 (0.1) | 14.6 (0.9) 11.1 (0.5) 9.6 (0.2) | 236 (12) |
| 13-20 | 16.6 | 12.3 | 4.3 | 2.6 | 483.5 (108.7) | 1.5 (0.1) | 11.5 (1.16) | 220 (29) |
| 20-23 | 5.4 | 4.6 | 0.8 | 2 | 850 (34) 1260 (241) | 1.1 (0.1) | 16.4 (0.6) | 95 (16) |
| 3-7* | | | | 2.6 | 202.3 (146.9) | 1.8 (0.4) | 10 (2.7) | 203.4 (11.9) |
| 13-20* | 15.2 | 10.5 | 4.68 | 2.6 | 486.5 (69.8) | 1.5 (0.1) | 11 (0.86) | 204 (2.5) |
| 13-20 | 15.7 | 11.3 | 4.4 | 2.6 | 510 | 1.47 | 12 | 204 |
| 20-26 | 6.1 | 5.1 | 1 | 2 | 1300 440 | 1.12 | 15.6 | 112 |
| 13-20* | 15.5 | 10.5 | 5.1 | 2.4 | 370 | 1.68 | 11 | 205 242 |



**Figure captions**

Figure 1. 50-m interpolated vertical profiles of aerosol concentration (cm$^{-3}$) with diameters 0.2-50 μm above Minorca Island (39.99°N, 4.25°E) **(a-b)** on 19 June at 1341 UTC and 28 June at 0530 UTC respectively, and above Ile du Levant (43.02°N, 6.46°E) **(c-d)** on 27 July at 2303 UTC and 3 August at 1056 UTC during the ChArMEx campaign in summer 2013.

Figure 2. **(a)** Latitude-longitude distribution of wind speed (ms$^{-1}$) at 300-hPa on 27 July at 1200 UTC (color scale: red relatively high values, blue relatively low values). Wind speed exceeds >50 ms$^{-1}$ above west of Portugal (yellow dashed line), solid lines indicate geopotential height (m) at 500-hPa derived from ECMWF operational analyses on 27 July at 1200 UTC, **(b)** Backward air mass trajectories using the FLEXTRA model starting at every 1-km height from levels of 15-25 km above Ile du Levant during 10 days from 27 July at 2300 UTC, interval between 2 triangles corresponds to a duration of 24 h. Color indicates altitude (km).

Figure 3. Filtered vertical profiles of temperature (°C), potential temperature (°K), zonal and meridional winds (ms$^{-1}$), ozone, LOAC aerosol concentration (cm$^{-3}$) of 0.2-0.7 μm size classes **(a, b, c, g, h, i)** and perturbations **(d, e, f, j, k, l)** on 27 July at 2303 UTC above Ile du Levant (43.02°N, 6.46°E). Solid blue (red) line visualizes filtered (background) profile. The prime symbol (') means perturbations.

Figure 4. Morlet CWT (given in units of physical unit$^2$ km$^{-1}$) on the left panels of perturbations of **(a)** normalized potential temperature, **(b) (c)** horizontal wind components u and v (ms$^{-1}$), respectively, and **(d)** ozone (ppmv) and **(e)** aerosol concentration (cm$^{-3}$) of size 0.2-0.7 μm size classes. Perturbations are visualized on the right panels. Color scale: red relatively high values, blue relatively low values. **(f)** Normalized fast Fourier transform of normalized potential temperature, normalized temperature, horizontal wind (ms$^{-1}$) and ozone (ppmv) and aerosol concentration (cm$^{-3}$) perturbations at heights of 15-23 km.

Figure 5. Vertical velocity (Pa s$^{-1}$) at 50 hPa in the lower stratosphere derived from ECMWF analyses on 27 July 2013 at 1800 UTC. The star symbol locates Ile du Levant. **(b)** Spectral density (Pa$^2$ s$^{-2}$ km$^{-1}$) as a function of latitudes for horizontal wavelengths (km) in east-west direction λ$_x$ <500 km, 500-1000 km and 1000-2000 km, **(c)** spectral density distribution at latitudes ranged between 42°N and 52°N and **(d)** spectral density at the latitudes of 42-48°N and 46°N.



Figure 6. **(a)** Location of GPS RO COSMIC data on 26, 27, 28 and 29 July 2013 marked with x, +, triangle, square symbols respectively. The black dot locates Ile du Levant. Numbers near markers indicate hours (UTC) of profiles from which **(b)** spectral densities ($C^2\,km^{-1}$) of GPS RO temperature perturbations (°C) are calculated for modes with vertical wavelengths < 5 km at altitudes of 10-20 km at heights of 10-20 km at longitudes of 2.5°-6°E and latitudes of 40-50°N on 26, 27, 28 and 29 July 2013 (upper panel to lower panel respectively).

Figure 7. **(a)** Vertical profiles of temperature (°C) and perturbations (°C) on the left and right panels respectively from RS and RO data on 27 July at 2303, 28 July 2013 at 0200 and 1200 UTC respectively. Morlet CWT (left panel) of temperature perturbations (right panel) of RO data on 28 July 2013 at **(b)** 0200 UTC and **(c)** 1200 UTC respectively. **(d)** Cross wavelet spectrum ($C^4\,km^{-2}$) on 28 July 2013 at 0200 UTC and 1200 UTC and corresponding horizontal wavelength (km) as a function of altitudes for vertical wavelengths between 2.5 km (blue solid line) and 3 km (red solid line). Color scale: red relatively high values, blue relatively low values.

Figure 8. **(a)** Observed (blue solid line) and simulated temperature (green solid line) in units of °C (left panel) respectively on 27 July at 2303 UTC. Same as the left panel but for horizontal wind profiles ($ms^{-1}$). Red solid lines indicate the observed mean profile. **(b)** Simulated perturbation profiles using GW simplified linear wave polarization relations. U', V' and T' are zonal wind ($ms^{-1}$), meridional wind ($ms^{-1}$) and temperature (°C) perturbations respectively.

Figure 9. Backward rays from Ile du Levant launched at 19 km height on 28 July at 0000 UTC onto **(a)** georeferenced infrared GMS-3 image and **(b)** altitude-latitude cross section (color scale: red relatively high values, blue relatively low values) of zonal wind ($ms^{-1}$) at longitude of 0.0 E on 27 July at 1200 UTC. **(c)** Evolution of horizontal wavelength in units of km (solid line), ω/f (dash-dot line) and time in units of h (dashed line) as a function of altitude (km).

Figure 10. **(a)** Normalized perturbations of the temperature and the ozone mixing ratio for the vertical wavelength bandwidth of 1.2-4.8 km using DWT. Temperature perturbation (°C) profile has been multiplied by the factor R(z) **(b)** defined in section 3.6. **(c)** and **(d)** Morlet CWT (left panels) of normalized temperature and ozone concentration (ppmv) perturbations (right panels) on 28 July at 1330 UTC (color scale: red relatively high values, blue relatively low values). **(e)** Specific humidity ($gkg^{-1}$) and perturbations on 27 July at 2303 UTC and 28




July at 1330 UTC at heights of 3-11 km, and **(f)** corresponding normalized spectral density of specific humidity.

Figure 11. Vertical profiles of **(a)** aerosol concentration ($cm^{-3}$) and **(b)** aerosol (size class: 0.2-50 µm) perturbations with 1.6-3.2 km vertical wavelengths and simulated vertical wind perturbations ($ms^{-1}$) on 27 July at 2303 UTC. On left panel, black (red) solid line corresponds to raw (background) profile ($cm^{-3}$). The sum of the background profile and GW perturbations is drawn as green solid line.





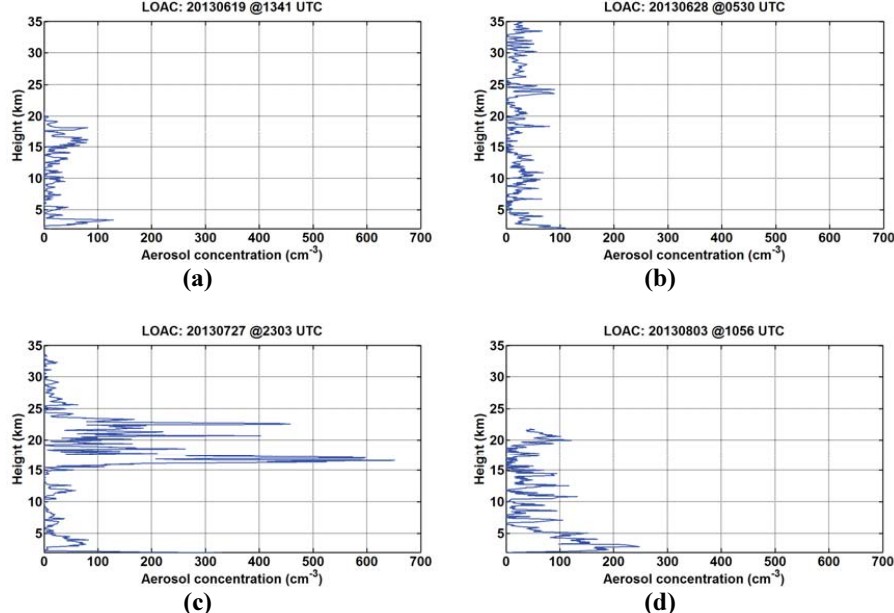

Figure 1. 50-m interpolated vertical profiles of aerosol concentration (cm$^{-3}$) with diameters 0.2-50 μm above Minorca Island (39.99°N, 4.25°E) **(a-b)** on 19 June at 1341 UTC and 28 June at 0530 UTC respectively, and above Ile du Levant (43.02°N, 6.46°E) **(c-d)** on 27 July at 2303 UTC and 3 August at 1056 UTC during the ChArMEx campaign in summer 2013.





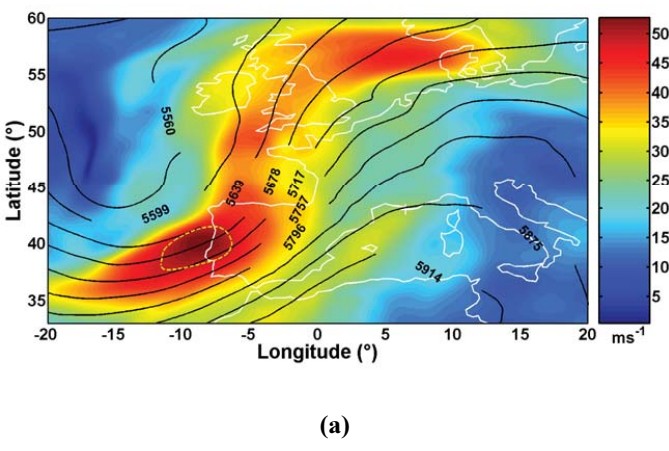

**(a)**

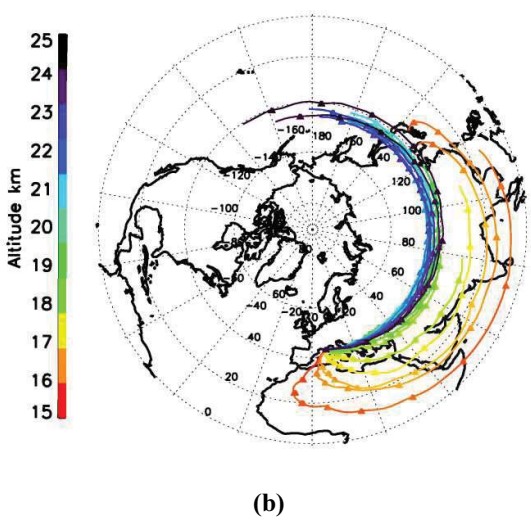

**(b)**

Figure 2. **(a)** Latitude-longitude distribution of wind speed (ms$^{-1}$) at 300-hPa on 27 July at 1200 UTC (color scale: red relatively high values, blue relatively low values). Wind speed exceeds >50 ms$^{-1}$ above west of Portugal (yellow dashed line), solid lines indicate geopotential height (m) at 500-hPa derived from ECMWF operational analyses on 27 July at 1200 UTC, **(b)** Backward air mass trajectories using the FLEXTRA model starting at every 1-km height from levels of 15-25 km above Ile du Levant during 10 days from 27 July at 2300 UTC, interval between 2 triangles corresponds to a duration of 24 h. Color indicates altitude (km).



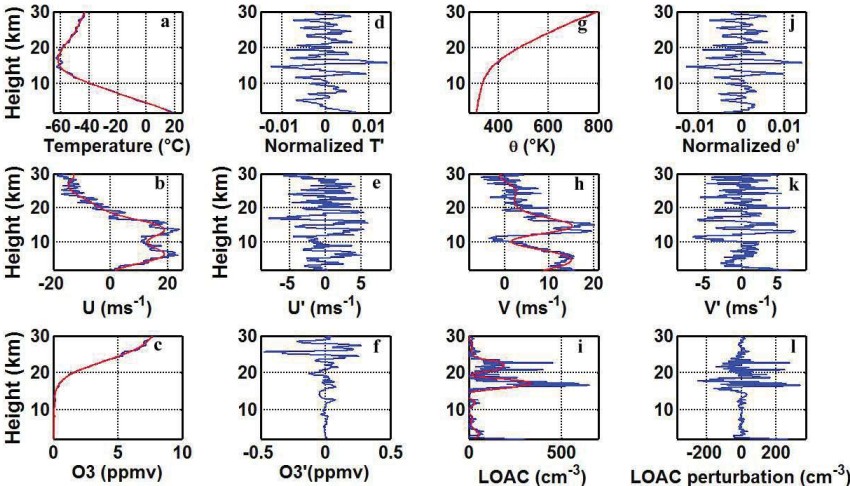

Figure 3. Filtered vertical profiles of temperature (°C), potential temperature (°K), zonal and meridional winds (ms$^{-1}$), ozone, LOAC aerosol concentration (cm$^{-3}$) of 0.2-0.7 μm size classes **(a, b, c, g, h, i)** and perturbations **(d, e, f, j, k, l)** on 27 July at 2303 UTC above Ile du Levant (43.02°N, 6.46°E). Solid blue (red) line visualizes filtered (background) profile. The prime symbol (') means perturbations.





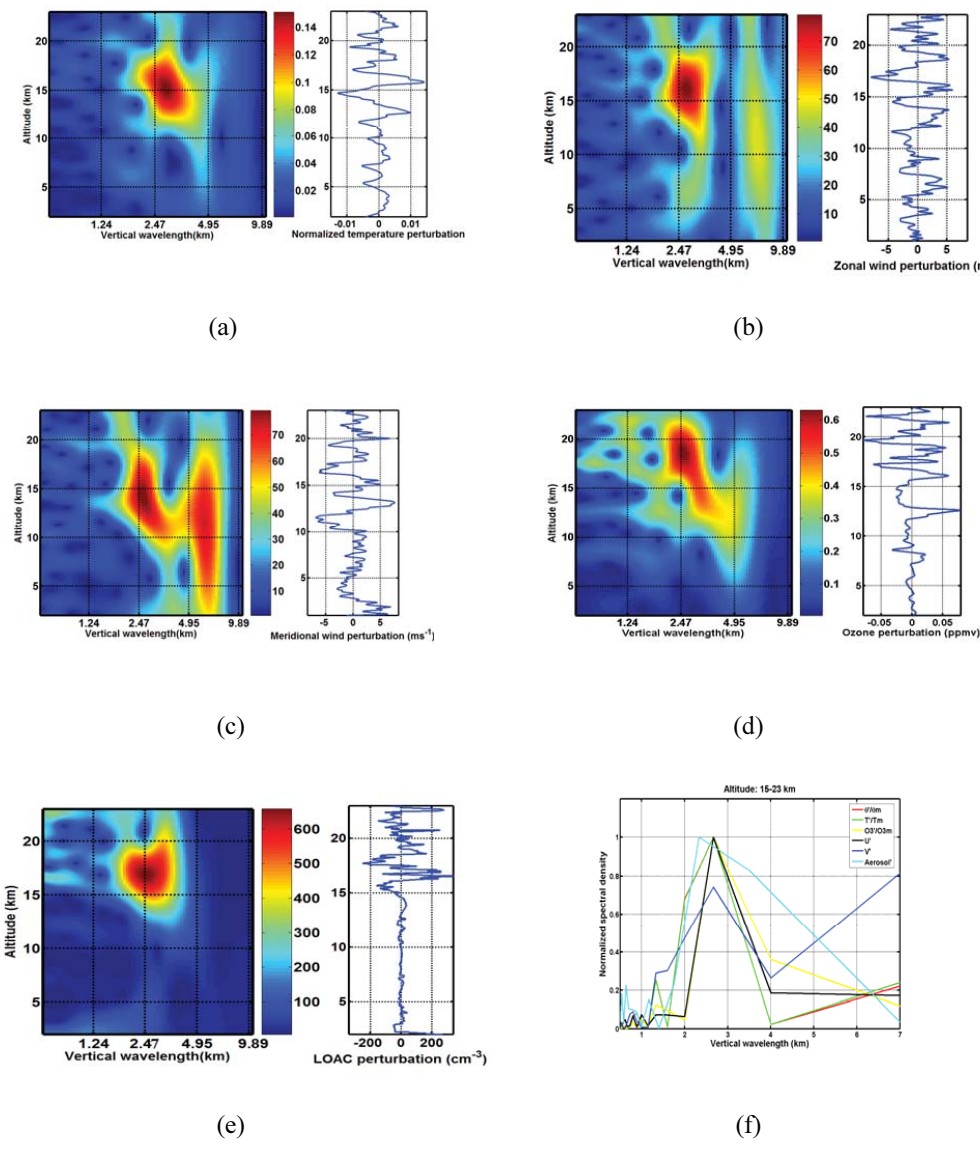

Figure 4. Morlet CWT (given in units of physical unit$^2$ km$^{-1}$) on the left panels of perturbations of **(a)** normalized potential temperature, **(b) (c)** horizontal wind components u and v (ms$^{-1}$), respectively, and **(d)** ozone (ppmv) and **(e)** aerosol concentration (cm$^{-3}$) of size 0.2-0.7 μm size classes. Perturbations are visualized on the right panels. Color scale: red relatively high values, blue relatively low values. **(f)** Normalized fast Fourier transform of normalized potential temperature, normalized temperature, horizontal wind (ms$^{-1}$) and ozone (ppmv) and aerosol concentration (cm$^{-3}$) perturbations at heights of 15-23 km.





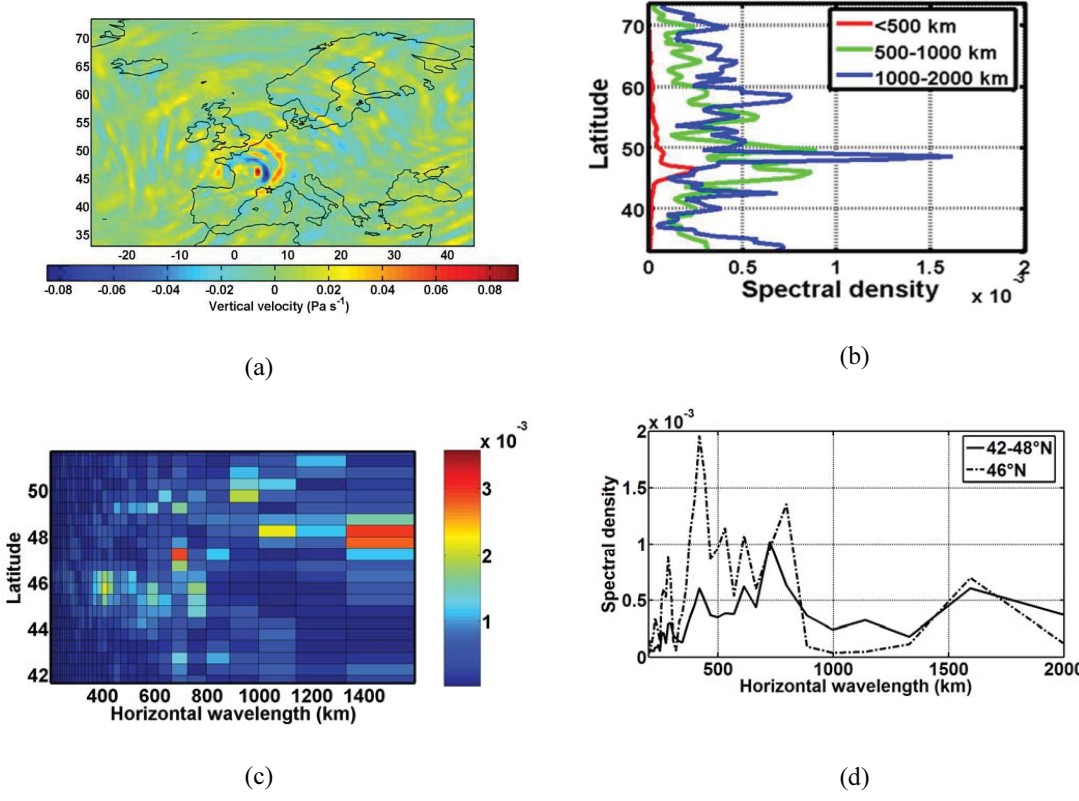

Figure 5. **(a)** Vertical velocity (Pa s$^{-1}$) at 50 hPa in the lower stratosphere derived from ECMWF analyses on 27 July 2013 at 1800 UTC. The star symbol locates Ile du Levant. **(b)** Spectral density (Pa$^2$ s$^{-2}$ km$^{-1}$) as a function of latitudes for horizontal wavelengths (km) in east-west direction $\lambda_x$ <500 km, 500-1000 km and 1000-2000 km, **(c)** spectral density distribution at latitudes ranged between 42°N and 52°N and **(d)** spectral density at the latitudes of 42-48°N and 46°N.





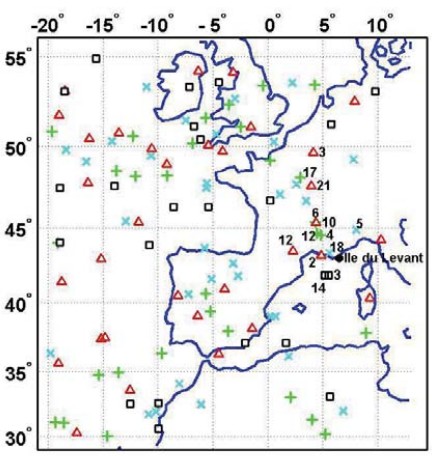

(a)

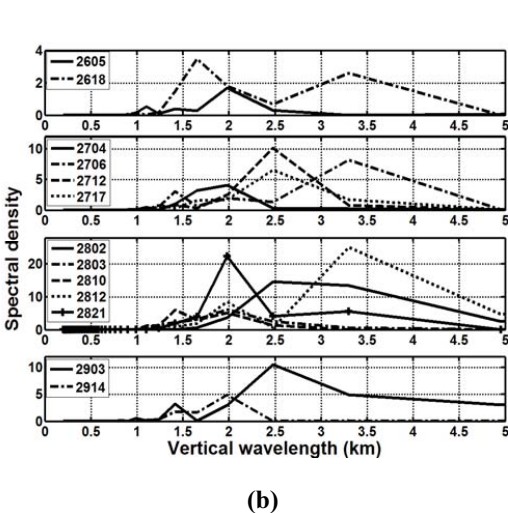

(b)

Figure 6. **(a)** Location of GPS RO COSMIC data on 26, 27, 28 and 29 July 2013 marked with x, +, triangle, square symbols respectively. The black dot locates Ile du Levant. Numbers near markers indicate hours (UTC) of profiles from which **(b)** spectral densities ($C^2\,km^{-1}$) of GPS RO temperature perturbations (°C) are calculated for modes with vertical wavelengths < 5 km at altitudes of 10-20 km at heights of 10-20 km at longitudes of 2.5°-6°E and latitudes of 40-50°N on 26, 27, 28 and 29 July 2013 (upper panel to lower panel respectively).



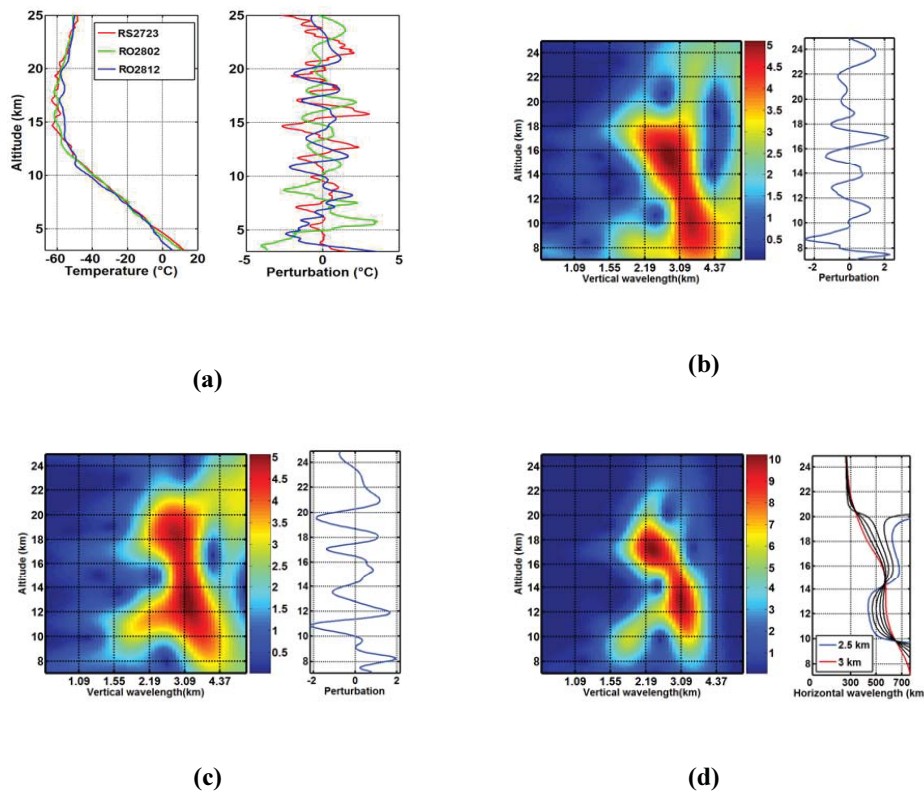

Figure 7. **(a)** Vertical profiles of temperature (°C) and perturbations (°C) on the left and right panels respectively from RS and RO data on 27 July at 2303, 28 July 2013 at 0200 and 1200 UTC respectively. Morlet CWT (left panel) of temperature perturbations (right panel) of RO data on 28 July 2013 at **(b)** 0200 UTC and **(c)** 1200 UTC respectively. **(d)** Cross wavelet spectrum ($C^4$ $km^{-2}$) on 28 July 2013 at 0200 UTC and 1200 UTC and corresponding horizontal wavelength (km) as a function of altitudes for vertical wavelengths between 2.5 km (blue solid line) and 3 km (red solid line). Color scale: red relatively high values, blue relatively low values.





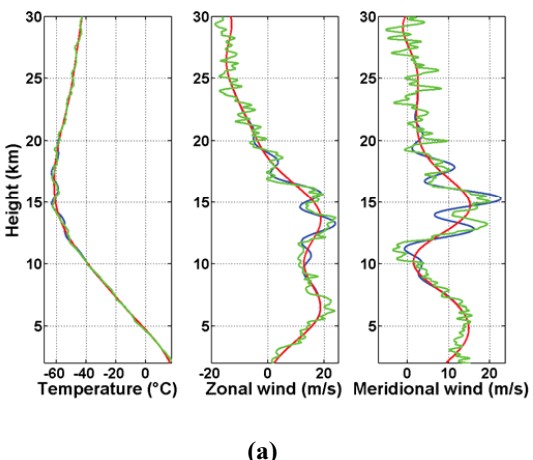

**(a)**

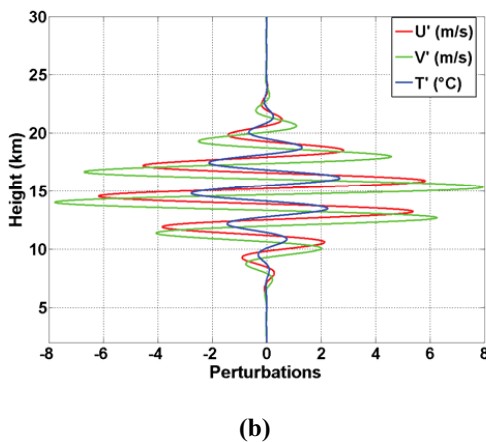

**(b)**

Figure 8. **(a)** Observed (blue solid line) and simulated temperature (green solid line) in units of °C (left panel) respectively on 27 July at 2303 UTC. Same as the left panel but for horizontal wind profiles (ms$^{-1}$). Red solid lines indicate the observed mean profile. **(b)** Simulated perturbation profiles using GW simplified linear wave polarization relations. U', V' and T' are zonal wind (ms$^{-1}$), meridional wind (ms$^{-1}$) and temperature (°C) perturbations respectively.



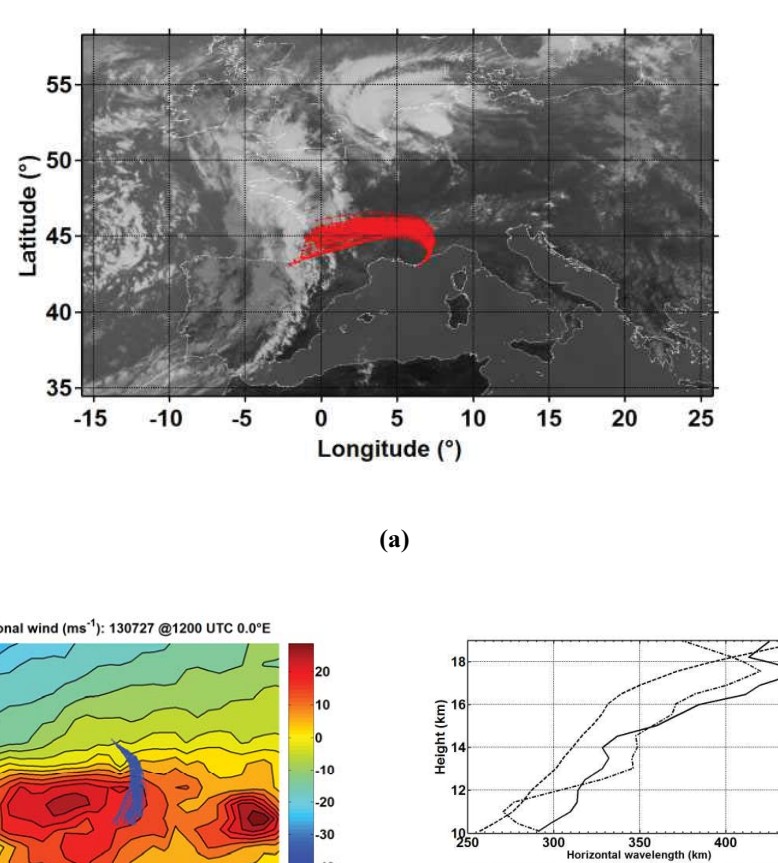

**(a)**

**(b)**

**(c)**

Figure 9. Backward rays from Ile du Levant launched at 19 km height on 28 July at 0000 UTC onto **(a)** georeferenced infrared GMS-3 image and **(b)** altitude-latitude cross section (color scale: red relatively high values, blue relatively low values) of zonal wind (ms$^{-1}$) at longitude of 0.0 E on 27 July at 1200 UTC. **(c)** Evolution of horizontal wavelength in units of km (solid line), ω/f (dash-dot line) and time in units of h (dashed line) as a function of altitude (km).



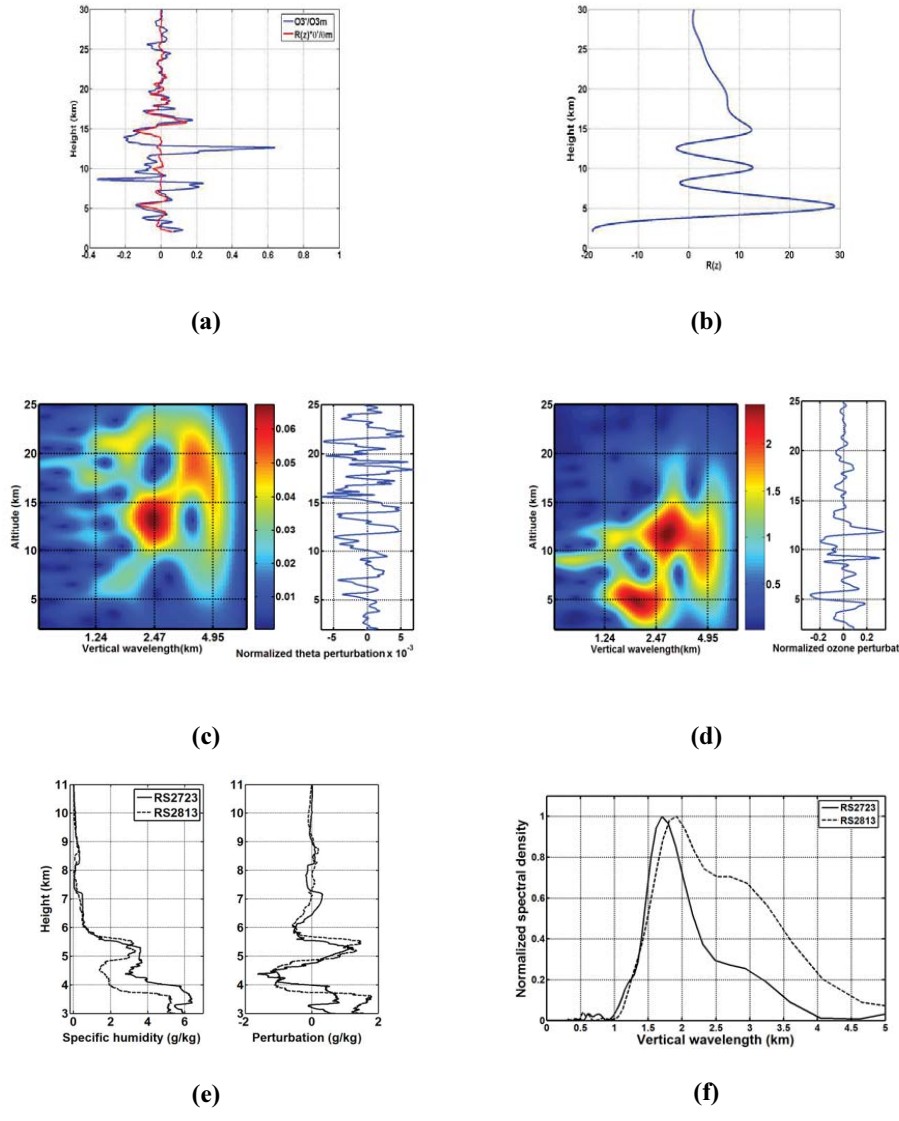

Figure 10. **(a)** Normalized perturbations of the temperature and the ozone mixing ratio for the vertical wavelength bandwidth of 1.2-4.8 km using DWT. Temperature perturbation (°C) profile has been multiplied by the factor R(z) **(b)** defined in section 3.6. **(c)** and **(d)** Morlet CWT (left panels) of normalized temperature and ozone concentration (ppmv) perturbations (right panels) on 28 July at 1330 UTC (color scale: red relatively high values, blue relatively low values). **(e)** Specific humidity (gkg$^{-1}$) and perturbations on 27 July at 2303 UTC and 28 July at 1330 UTC at heights of 3-11 km, and **(f)** corresponding normalized spectral density of specific humidity.



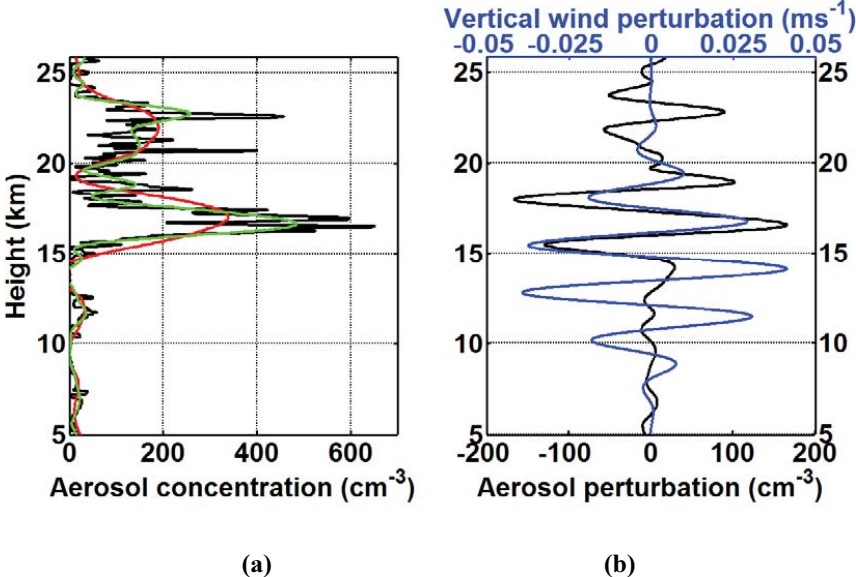

Figure 11. Vertical profiles of **(a)** aerosol concentration (cm$^{-3}$) and **(b)** aerosol (size class: 0.2-50 μm) perturbations with 1.6-3.2 km vertical wavelengths and simulated vertical wind perturbations (ms$^{-1}$) on 27 July at 2303 UTC. On left panel, black (red) solid line corresponds to raw (background) profile (cm$^{-3}$). The sum of the background profile and GW perturbations is drawn as green solid line.