# Peer review of "Gravity-wave effects on tracer gases and stratospheric aerosol concentrations during the 2013 ChArMEx campaign"

_Atmospheric Chemistry and Physics, 2015_

## Referee Comment (RC1) · Anonymous Referee #3 · 26 Feb 2016

**General Comment**

In their paper, the authors address the effect of mesoscale gravity waves on the distribution of aerosol and ozone in the Mediterranean area during July 2013. Vertical profiles of balloon-borne Light Optical Aerosol Counter (LOAC), M10 meteorological global positioning system (GPS) sondes, ozonesondes and GPS radio occultations are analyzed to identify fluctuations caused by gravity waves (GWs). By applying different techniques GW characteristics are derived, and by backward ray-tracing a jet-front system is identified as the source of the observed GWs. Finally, the phase relationship between GW induced fluctuations in ozone, aerosol concentrations and wind perturba-

tions is discussed.

The effect of gravity waves on trace gas or aerosol distributions is a topic that has seldomly been studied and is therefore of relevance for readers in the fields of both atmospheric chemistry and atmospheric physics. There are, however, several concerns, two of them major, that should be addressed before publication of the paper in ACP.

**Major Comments**

(MC1) Multiple waves are identified in Table 1. For the case discussed, this would be expected because there is a strong wind reversal at 18 km altitude, and the wave parameters in Table 1 show considerable spread.
Nevertheless, when reading the text, I had the impression that sometimes while writing, you were thinking of just one single wave being detected. Therefore it is not always very clear and/or consistent which of those waves you are discussing.

This is most evident in Sect. 5.3 where you attribute wave parameters over a large altitude range to just a single wave, which obviously is not correct.
In addition, the discussion of the backward ray traces also somehow suffers from this issue.

Regarding this issue, please carefully check the whole manuscript for clarity and consistency.

(MC2) For the three RS and RO temperature profiles presented, the phase difference method cannot be applied to extract wave properties because the time differences are way too large! Therefore I would suggest to either select other profiles, if possible, or to just drop the related discussion because it is not necessarily needed.

Please find more detailed comments below.

**Detailed Comments**

(1) p.2, l.30+31: here you write

"...are now capable to capture some characteristics of GWs using proper GW parameterization..."

This statement is somehow misleading. By parameterization only the "effect" of GWs is captured, not the "characteristics" because the waves are not resolved

Nevertheless, a certain portion of the GW spectrum will indeed be resolved by global models. However, by comparison with observations, it has been shown that the resolved GWs are usually under-represented (Schroeder et al., 2009). This is somehow mentioned later in the text in l.32–34, but should be more clearly stated.

Reference:
Schroeder, S., Preusse, P., Ern, M., and Riese, M.: Gravity waves resolved in ECMWF and measured by SABER, Geophys. Res. Lett., 36, L10805, doi:10.1029/2008GL037054, 2009.

(2) p.2, l.31+32:
Please check! I don't think Shutts and Vosper (2011) mention that convection would be resolved!
Just the opposite is true for the two NWP models they discuss. Convection in the MetOffice model and in the ECMWF IFS is parametrized (their Sects. 2.1 and 2.2), and in their Sect. 3.1, they state:
"Note that, since convection is parametrized in both models, it seems unlikely that the amplitude of these waves will be well represented.",
and later in their Sect.6:
"tropical gravity wave activity is under-represented in the model-derived gravity wave fields since parametrized convection is a rather ineffective forcing agency."

(3) p.5, l.28 onward:

Since you are using RO (wet) temperatures at altitudes as low as 3km, are there significant differences between dry and wet temperatures?

In the lower troposphere the effect of water vapor on GPS bending angles becomes increasingly important. Therefore the bending angle signal caused by water vapor fluctuations could map into temperatures and could somehow bias the GW signal in the temperatures. (Of course, GW fluctuations in both wet and dry temperatures would be biased because usually for wet temperatures water vapor is taken just from reanalysis or climatology.)

(4) p.8, l.30 until end of Sect.3.3 — not entirely correct...

Horizontal and vertical wavelengths in Wang and Alexander (2010) or Faber et al. (2013) are global distributions of "average" values. Of course, their methods can also detect shorter scales, similar as in your study!

(5) p.8, l.26 until p.9, l.3:

For the three RS and RO temperature profiles presented, the phase difference method cannot be applied because the time differences are way too large! Therefore I would suggest to either select other profiles, if possible, or to drop this paragraph.

Still, the vertical wavelength information provided later in Fig.6 is quite useful.

(6) p.14, l.2: Here it is somehow unclear: Are you discussing just one dominant wave, or multiple waves, or different effects and one effect forgotten??!

waves are both filtered → the observed wave is filtered ??

Please clarify!

(7) caption of Fig.6 and elsewhere:

units like $C^2\,km^{-1}$ are not SI standard and even ambiguous, please either use

$^{\circ}C^2\,km^{-1}$ and similar, which is still not SI standard but OK, or better use $K^2\,km^{-1}$ and similar, throughout

(8) p.15, l.20:
Here, it should be more emphasized that obviously different waves are detected.

(9) p.15, l.28: Is this a strong event? How does this momentum flux compare with other findings?

For example, you should mention that in summer midlatitudes GW momentum fluxes are usually low in the stratosphere, as indicated by satellite observations (for example, below around 0.001 Pa at 25 km over Europe as shown in Ern and Preusse, 2012). Your value of $0.05\,m^2\,s^{-2}$ at altitudes 13–20km corresponds to about 0.008 Pa which is well beyond this value, even taking into account that the altitudes are somewhat different.

Reference:
Ern, M., and Preusse, P.: Gravity wave momentum flux spectra observed from satellite in the summertime subtropics: Implications for global modeling, Geophys. Res. Lett., 39, L15810, doi:10.1029/2012GL052659, 2012.

During summer, GW momentum fluxes obtained from radiosondes over North America are about $0.02\,m^2\,s^{-2}$ around 17 km altitude (for example, Zhang et al., 2014). Usually, during summer momentum fluxes over North America are somewhat higher than over Europe. Nevertheless, your values over Europe are twice as high as those reported over North America. This further supports that your case represents a stronger GW event.

Reference:
Zhang, S. D., Huang, C. M., Huang, K. M., Yi, F., Zhang, Y. H., Gong, Y., and Gan, Q.: Spatial and seasonal variability of medium- and high-frequency gravity waves in the lower atmosphere revealed by US radiosonde data, Ann. Geophys., 32, 1129—1143, 2014.

ACPD

(10) p.15, l.31: Here you state: "which means an excess of wave energy near the inertial frequency" — This statement is too strong and needs some explanation!

Hertzog et al. (2002) mention that p should be in the range of about 1–2 (theoretically about 5/3), and they find values in the range 1.5–2.2 for the "standard" power-law part of their observations.
Your values of 2.6 to 2.9 are only somewhat outside this range and not necessarily something special. Further, your statement is based on the observation of just a few waves, and your analysis involves a certain error range.

Please note that Hertzog et al. (2002) mention values exceeding p=5 as quite high and not fully understood. I am not sure whether they would be worried about values p<3. They suggest that exceedingly high values could be caused by enhancements of the velocity spectrum near the inertial frequency. Indeed, they find enhancements in two out of three balloon flights. These enhancements over the power-law spectrum are, however, a factor of TEN, which is far beyond your values.

Therefore you should add some explanation about the expected range of p, and you should use a weaker statement.

(11) p.16, l.12 until end of Sect. 5.2:
The time difference between the different soundings of the "triad" is too large for deriving horizontal wavelengths by the phase difference method.

Several criteria have to be matched to obtain useful information of the horizontal wave structure by applying the phase difference method. A more detailed discussion is given in Schmidt et al., 2016.

Reference:
Schmidt, T., Alexander, P., de la Torre, A.: Stratospheric gravity wave momentum flux from radio occultations, J. Geophys. Res. Atmos., doi:10.1002/2015JD024135, in print, 2016.

One of the preconditions is that there should be no significant phase progression of the wave due to its frequency. Otherwise phase differences are no longer dominated by the horizontal wave structure. Schmidt et al. recommend time differences of no more than 15 minutes. The three soundings you are using, however, are spread over 13 hours. This time span is much too long compared to the wave periods in your case, which are in the range 10 to 16.4 hours, as listed in your Table 1.

I would suggest to either select other profiles, if possible, or to delete this whole part of the subsection.

(12) p.17, Sect.5.3: Here you state:
"The spectral parameters (a vertical wavelength of 2.6 km,...) at heights of 12.8 km with a height range of 14 km are used to produce synthetic RS profiles with the signature of such a dominant mesoscale GW..."

Obviously, here you assume there is just one single dominant wave in a whole range of 14 km altitude, and for this wave you claim:
"Because of the variation of the Brunt-Väisälä frequency with height, the horizontal wavelength, computed using the GW dispersion relation, has values of 225.3 $\pm$22.6 km and 499.9 $\pm$45.7 km at heights of 3–7 km and 13–20 km respectively."

This kind of approach is not correct!
Variations of the BV frequency in vertical direction will NOT influence the HORIZONTAL wavelength! They will only have effect on the VERTICAL wavelength! See the refraction equations (Eq. 2.12) in Olbers (1981):

$dk_i/dt = -\partial\Omega/\partial x_i$

Reference:
Olbers, D. J.: The propagation of internal waves in a geostrophic current, J. Phys. Oceanogr., 11, 1224–1233, 1981.

[Figure]

Similar reasoning regarding the GW raytracing equations is found in Marks and Eckermann (1995), Appendix A:

dk/dt = ... $(N^2)_x$ ...

dl/dt = ... $(N^2)_y$ ...

dm/dt = ... $(N^2)_z$ ...

where the subscript means the gradient in the respective direction (x,y,z).

Changes in the horizontal wavenumbers are not related to vertical gradients in the BV frequency. Different from this, assuming just one wave with a fixed vertical wavelength in this whole altitude interval containing the tropopause with a strong jump in the BV frequency and strong changes in the zonal wind of more than 20 m/s may be over-simplified!

Differences (or even unexpectedly unchanged values!) in wave parameters listed in Table 1 should be caused by observing different waves at different altitudes.

Sect. 5.3 with the simulated profiles should therefore be revised!

(13) p.17, l.27 and later:

Same issue: for a given GW, the horizontal wavelength should not change much.

Therefore statements like:

"The horizontal wavelength (period) decreases (increases) slowly from 440 km (10 h) at 17 km heights to 290 km (14 h) at 10 km heights."

need further explanation.

Is the wavelength mentioned here the wavelength of the GROGRAT component with maximum amplitude at a given place and time?

What does this change in wavelength mean? Is this just an effect of the raytracing technique and the selection of input parameters, or do you think this provides information about the GW distribution for the case discussed, for example where waves with a given combination of wave parameters could preferentially be observed?

As far as I understand, the main finding of your simulation is that almost all rays can be attributed to the frontal source, even if the exact wave parameters are not known. Therefore it should be more emphasized that waves with a whole range of parameters could be excited by the front and propagate to the location of Ile du Levant.

This part of Sect. 5.3 should also be revised accordingly.

(14) p.21, l.10/11: Again, here you write: "when a mesoscale inertia GW produced by the jet-front system was identified during a jet-streak event"

Again, it is misleading to talk of just a single wave! From your analysis, it rather looks like different waves are seen at different altitudes, or even at the same altitude.

**Other Comments**

- p.2, l.4+5:
  European Center for Medium range Weather Forecasting (ECMWF) → European Centre for Medium-Range Weather Forecasts (ECMWF)

- p.2, l.7: oscillations → oscillations of

- p.2, l.19+20:
  for momentum transport and deposition → by momentum transport and momentum deposition

- p.3, l.25: The microphysical → Their microphysical ??

- p.4, l.12: Results on → Results of

- p.4, l.22: The campaign occurred → The campaign was carried out

- p.5, l.4: 3000 particles → 3000 particles $cm^{-3}$ ??

- p.6, l.27: of iterated so that → of iterations such that ???

- p.12, l.13-16: Please check!
  According to the caption of Fig. 3, the aerosol class shown in Fig. 3 is 0.2–0.7 $\mu$m, and not 0.2-50 $\mu$m as stated in the manuscript on p. 12.

- p.13, l.21: enhances ozone peak → would enhance the ozone peak

- p.15, l.6: Figure 6b indicates that spectral → Figure 6b indicates spectral

- p.15, l.27: 16 J.kg$^{-1}$ → 16 J kg$^{-1}$

- p.21, l.22: lead → leads

- caption of Fig. 8: It looks like for (a) blue and green lines are interchanged in the caption. Please check!

---

## Referee Comment (RC2) · Anonymous Referee #2 · 31 Mar 2016

The paper by Chane-Ming et al. uses a number of well established technique for an in-detail case study on GWs observed above southern France. There is need on more detailed studies on the interaction of GWs with tracers, one main focus of this paper. Also, there is a discussion on aerosols. However, there are several major points which need to be addressed before the paper can be published in ACP.

Major comments:

1.) GW source:

There are a number of questions related to the source process. The wave properties from ECMWF are largely consistent with the observations. If the patterns in ECMWF

capture the truth, they can point to the source. Given the concentric nature of the GW patterns that would be a point source at 2.5E, 46N; a point which is also passed by the backward trajectories. The authors should investigate the vicinity of that point.

2.) Horizontal wavelength from triples

The only thing you can learn from that analysis is that the wavelength is mesoscale. I therefore recommend to move the analysis to an appendix and include only a few sentences in the body of the text. Details below.

3.) There are some inaccuracies in discussions and descriptions, see the specific comments below

4.) Language

I did not make many language comments, but the English needs to be improved. In addition it would be nice to have some introductory or explaining sentences why you do something and not only how.

5.) Conclusion A large part of the paper deals with the analysis and potential source of the GWs. This is not reflected in the conclusions which focus almost entirely on aerosol processes.

6.) Last but not least: Improve the figure quality!

I have noted so as a main point in my prereview and was disappointed to see that nothing changed. The single panels are mostly o.k., but combined they look awkward. Panels of the same format have different sizes. Color bars are sometimes at the side, sometimes below the plot - have it one way if you combine in one figure!

6b) and consistency of the maps

You show the same region of Europe with four different quantities (horizontal wind, vertical wind/GWs, GPS-RO measurements, clouds and trajectories) and you use as many different map limits! It would make comparison much easier to use the same!

Specific comments

P7L11 Average of what?

P9L1 In the present study, RS and RO temperature ... The described method is based on the assumption that the same wave is observed in several profiles. Since the limit is given by the visibility filter of the instrument, you cannot beat the shorter limit of these previous study.

P9L28 This is inaccurate: GROGRAT has quite a large number of criteria to stop a ray-trace. When you use backtracing the following may be source indicative: There are a number of criteria in GROGRAT which at the end all apply to the ray approaching a critical level. In that case the *wavenumber*, i.e. the inverse of the wavelength approaches inifinity. In forward raytracing this just denotes a critical level which the GW cannot pass and below which it deposits its momentum. In backward raytracing the wave must origin above the critical level and wind shear is likely involved in the wave excitation (cf. also discussions in Preusse et al. 2014, Pramitha et al., 2015). GROGRAT also monitors, whether the WKB assumption that the background is changing slowly compared to the vertical wavelength is still valid. A WKB violation may be indicative of a source due to spontaneous imbalance (cf. Hertzog et al., 2001). Whether GROGRAT terminates the ray calculation at a point of WKB violation actually is the choice of the user. A vertical wavelength approaching infinity would point to total reflection (or, at start, to an evanescent wave) which however I would not associate with a source process. There is no limit to the rate with which the vertical wavelength changes. Please reformulate!

P11L9 better such features play

P11L13 severely impact on severe I would more connect with the impact of the weather on people and property -> have large influence on (control would be stronger but that is, I think, more than the authors found and which was more like a seeding process)

P11LL29/Figure2: Please motivate why you combine winds at 300hPa with geopotential at 500hPa.

Figure 4: The panel labels should be at the upper left. The individual panels, though using the same format, are of different size. I had commented on this in the prereview - it is not really much work to improve this and it indicates some carelessness inappropriate to a scientific publication in a high level journal such as ACP.

P14L6 What precisely is in quadrature? Temperature and winds along the main axis should be ... For the winds it should depend on the orientation of the wave vector / main axis, but that is determined from the winds?. Please explain.

P14LL20 The waves have nowhere plane wave fronts. They have a close-to-semicircle shape. Accordingly, the wavelengths projected to the west-east direction become (seemingly) longer the further you move away from the circle center. Just on looking at the picture, in 2D the wavelengths are pretty similar for all latitudes. Though 5c is somewhat scattered in principle one can see that effect.

Fig 5b: I guess you calculate this from F5c? By averaging over the spectral bins? If yes, I think integrating would be preferable for the following reasons: According to Parseval theorem the integral of the variance is the integral over all spectral components. FT than distributes the variance to a larger number of spectral bins for the shorter wavelengths which provide better spectral resolution. In that sense an integral would be more adequate as it represents the total variance in that wavelength range. Use an integral would hence less overemphasize the long horizontal wavelengths. In addition, from F5a the largest total variance is clearly at 46N. Now it appears to be at 48N. The really interesting variance in the red curve is almost disappearing.

Fig 5d The peaks in the spectra seem to be at 400km, 800km and 1600km. This could potentially suggest that you see a main wavelength and the subharmonics due to the fact that the amplitudes varies spatially.

Figure 6a: The map shows a much larger region than needed for the discussion.

P15 first paragraph: The correspondence of figures and text should be improved. F6a shows the location of the profiles. The position of the bracket in the text suggests it is the wavelength. That is then shown in F6b again with bad reference. ... And so it goes on!

Why not something like:

The waves from this event can be identified in GPS-RO soundings as well. Figure 6a shows an overview of the GPS-RO soundings over western Europe for the days 26 to 29 July, 2013. We selected profiles for longitudes of 2.5°-6°E and latitudes of 40-50°N for spectral analysis of the altitude range xx-yy km. The results are shown in Figure 6b. The individual spectra are labelled by the day and UTC of the measurement (e.g. 2605 for 26 July; 05 UTC) and marked also accordingly in Figure 6a. As can be seen from the consecutive days, GWs are enhanced starting from 27 July, peak at 28 July and are still active on 29 July. Intensity peaks are found for wavelengths 2-3.5 km.

Still no clue for which altitude range this is, but I hope for the lower stratosphere. A paper should not be a puzzle!

Also the following lines: You can't say anything about horizontal structures from the observations yet!

Table 1: Why do you have two/three values of horizontal wavelength for the same vertical wavelength? Please include in table caption!

P16LL12 You describe in the text that you have quite a number of reasons for uncertainty: a) close to Nyquist b) additional phase shift by radiative transfer / retrieval c) temporal development as the profiles are not measured simultaneously (needs to be included)

If you had very long wavelengths, you could identify them using the method, everything else is not sure. I recommend to move the details of the analysis (P16L12 to P16L33) into an appendix and include into the text only a few lines such as:

We have also combined radisonde and GPS-RO profiles to triples and applied the phase difference method of Faber et al (2013) to estimate the horizontal wavelength (see appendix 1 for details). We result in a horizontal wavelength of 400km consistent to ECMWF results, but any other (in particular shorter) mesoscale wavelength such as found from the radiosondes analysis (Table 1) would be also compatible inside the error range. Since the horizontal wavelength is likely overestimated, a GWMF estimate of 0.03 m2/s2 is somewhat lower but compatible with the estimate from the radiosondes alone, which is 0.05 m2/s2.

Move P15L9-L13 to the conclusion of this paragraph.

Page 18L1: GROGRAT adjusts the time step according to an internal accuracy estimate. Omit: "and a time step of 6 min."

P18 first paragraph: I have questions and, overall, I don't agree with your conclusion.

First to the questions: All your backward trajectories seem to end at the same altitude. To my experience in a well-setup simulation the termination altitude varies when the input parameters are varied. What is the termination condition? GROGRAT provides it!

If it were WKB: are you sure it is not simply because of the tropopause? The tropopause is a steep gradient in N, which almost always causes some WKB violation, if adequately resolved. It is in this case not indicative of a source.

Second to the comment: The source could be also above, somewhere along the ray trajectories. All trajectories pass close to 2.5E, 46N. This location seems the origin of the semi-circle (maybe even concentric, just weaker to the west) wave patterns in ECMWF. The values from the radiosondes seem to be consistent with ECMWF: southeastward propagating waves with mesoscale wavelengths: At least the direction if not the wavelength is consistent with the circular wave patterns.

If we believe in the circular wave patterns, this shines a different light on the wave

source mechanism, though. There are two kind of waves associated with jets and fronts: 1. GWs from spontaneous adjustment are generated in the jet-exit region and wave fronts are roughly perpendicular to the main flow. They would be organized in bow-shaped structures but not in circles. 2. Waves from the front have wave fronts parallel to the front and parallel to the main wind. Both don't cause circles. Circular patterns are much more likely to be generated by e.g. thunderstorms.

You reported thunderstorms in the beginning of the paper. Was there a thunderstorm close to 2.5E, 46N?

---

## Author Comment (AC1) · 11 May 2016

Detailed reply to comments from the two anonymous referees of manuscript "Gravity-wave effects on tracer gases and stratospheric aerosol concentrations during the 2013 ChArMEx campaign" (acp-2015-889)

Fabrice Chane Ming1, Damien Vignelles2, Fabrice Jegou2, Gwenael Berthet2, Jean-Batiste Renard2, François Gheusi3 and Yuriy Kuleshov4,5,6,7

1Université de la Réunion, Laboratoire de l'Atmosphère et des Cyclones, UMR 8105, UMR CNRS-Météo France-Université, La Réunion, France 2CNRS, LPC2E, UMR 7328, CNRS / Université d'Orléans, Orléans, France 3Laboratoire d'Aérologie,

UMR5560, Université de Toulouse and CNRS, Toulouse, France 4Bureau of Meteorology, Australia 5School of Science, RMIT University, Australia 6Faculty of Science, Engineering and Technology, Swinburne University, Australia 7School of Mathematics and Statistics, Melbourne University, Australia

First, we thank the anonymous reviewers for their helpful detailed comments and suggestions on our manuscript. In the following, comments of the reviewers are fully addressed and modifications have been made in the revised manuscript accordingly. Responses to comments of referee #2 are highlighted in red; those of referee #3 in blue as well as revised sentences and paragraph in the updated manuscript. A new version of the manuscript is attached as supplement.

Yours sincerely, Fabrice Chane Ming

Referee 3:

Major Comments

(MC1) The whole manuscript has been checked for clarity and consistency about discussion on multiple waves as it is supported by the spread of parameters in table 1 (horizontal wavelengths and $\omega$/f) . Indeed, our methods of analysis of GW characteristics (Table 1) are focused on dominant GWs among wave packets. Modifications in the text are presented in our responses to specific comments.

(MC2) This part has been modified. The phase shift is now computed between RS2723 and RO2802 for which time variation and the distance are 1.8h and 179.14 km respectively. Because time variation is > 15 minutes, our calculation of phase shift takes into account the time variation using the estimated $\omega$ (Table 1). The authors think that it is important to report the estimation of wave parameter from GPS RO measurements because they also support observations of GWs (refer to comment 5 for more details).

Specific Comments

Comment (1 & 2): Response: The text has been modified as suggested by the reviewer. Change P2L31: Thanks to recent progress in computer technologies, current operational numerical weather prediction models have sufficient spatial and temporal resolution to resolve the portion of GW wave spectrum with horizontal wavelengths of 100-1000 km (Shutts and Vosper, 2011). However, global climate models as well as numerical weather prediction models still need a set of GW parameterizations with a large number of tunable parameters for a realistic representation of the middle atmosphere (Preusse et al., 2014). By comparison with observations, it has been shown that the resolved GWs are usually under-represented (Schroeder et al., 2009).

Comment (3): Response: The authors have compared GPS-RO dry and wet temperature profiles used in this study. They are similar above the altitude of 10 km. We mentioned that our study is focused on heights of 10-20 km because GW fluctuations in both wet and dry temperatures might be biased below 10 km heights.

Change P6L4: We have removed 'wet' from the text and included the following sentence "Because GW fluctuations in both wet and dry temperature profiles would be biased by the effect of water vapor at heights below 10 km, the study is focused on heights above 10 km.

Comment (4): Response: The authors have removed the last sentence at the end of Sect.3.3 and modified the text Change P9L3: "At a given altitude, the horizontal wavelength can be deduced from adjacent vertical profiles of temperature close in time and space in order to observe the same GW packet. Thus, the time variation in the phase difference can be neglected (refer to equation 5 in Wang and Alexander (2010)) and the phase shift divided by the distance provides the horizontal wavenumber projected along the line connecting the two profiles. Ern et al. (2004) introduced this method to estimate horizontal wavelength of GWs and global absolute values of vertical flux of horizontal momentum at 25 km altitude from adjacent temperature profiles from Cryogenic Infrared Spectrometers and Telescopes for the Atmosphere (CRISTA). The method is adapted to pairs and triads of RO temperature profiles using the S-transform and CWT in the altitude range of 17.5–22.5 km with temporal windows of 4 h

and 2 h (Wang and Alexander, 2010; Faber et al., 2013). To better constrain estimated horizontal wavelengths and momentum fluxes, Schmidt et al. (2016) used temporal and spatial windows of 250 km and 15 minutes."

Comment (5): Response: The authors have modified the text and only compute the phase shift between RS2723 and RO2802 at the altitude of 17 km for which time difference is 1.8 h because of the ascent time of the RS. In addition, the calculation of the horizontal wavelength takes into account the time variation by using the value of period and the uncertainty of the direction of horizontal propagation (Phi) reported in Table 1. Fig. 7d has been removed. Scalograms of RO2802 and RO2812 have been preserved because they complete Fig6b. Indeed they support the presence of GWs with 2-3 km vertical wavelength in the troposphere and the lower stratosphere on 28 July like observations on RS2723.

Change P16L28: the text has been modified as follows: "Figure 7a visualizes temperature profiles from RS at 2303 UTC (hereafter called RS2723) on 27 July above Ile du Levant and RO at 0200 UTC (hereafter called RO2802) and 1200 UTC (hereafter called RO2812) on 28 July. As observed on the RS2723 temperature profile, GPS RO temperature profiles also show evidence of small-scale perturbations in the troposphere and the lower stratosphere. Scalograms of RO2802 and RO2812 temperature perturbations support the presence of dominant GW structures with vertical wavelengths of 2.5-3 km at heights of 10-18 km on 28 July (Fig. 7b, c). By assuming that the same GW packet is observed on RS2723 and RO2802 profiles in the LS, the phase shift () between perturbation profiles is calculated at the altitude of 17 km taking into account of the time variation (refer to section 3.3) using a GW period of 12 h at heights of 15-20 km and a time difference of 1.8 h at the altitude of 17 km between RS and RO measurements. Using a distance of 179.14 km between temperature profiles, the phase shifts () of 1.67 radians provide an horizontal wavelength of 673.6 km. The estimated horizontal wavelength is larger than the 'real' value by a factor $1/\cos \alpha$, where $\alpha$ is the angle between the connecting line of the two profiles and the real horizontal

wave vector (Preusse et al., 2002). Thus the 'real' horizontal wavelength is ranged between 396 and 674 km (Phi=200±29°). The result is consistent with values of horizontal wavelengths derived from applying conventional methods on RS2723 profiles at heights of 13-20 km."

Comment (6): Response: Here we discuss multiple waves characterized by a dominant wavelike structure (peak of intensity). The word "both" has been removed.

Comment (7): Response: We have done the modification "°C2 km-1"

Comment (8): Response: Modifications have been done. Change P15L27: "In particular, the hodograph analysis reveals the presence of mesoscale inertia GWs"

Comment (9): Response: We have included suggestions of the reviewer concerning comparison with previous studies in summer mid latitudes. Change P16L5: "Our computed value of vertical flux of horizontal momentum (about 8 mPa) is well beyond values of 1 mPa and 0.02 m-2s-2 observed in the LS in summer midlatitudes respectively by Ern and Preusse (2012) over Europe from High Resolution Dynamics Limb Sounder (HIRDLS) observations and Zhang et al. (2014) from radiosondes over North America. Thus the value of vertical flux of horizontal momentum supports our statement that the case on 27 July 2013 represents a stronger GW event."

Comment (10): Response: We have added some explanation about the expected range of p and modified our statement. Change P16L10:" The ratio between kinetic and potential energy of GWs provides a spectral index (p) of about 2.6-2.9 which is larger than the theoretical values of p (about 5/3). However, Hertzog et al. (2002) find values of p in the range 1.5-2.2 for high-frequency GWs from superpressure balloon measurements in the stratosphere. They suggest that values greater than 5 could be caused by enhancements of the velocity spectrum near the inertial frequency."

Comment (11): Response: Refer to our response to comment (5). We provided a reference on paper of Schmidt and Alexander (2016) at the end of section 3.3.

Comment (12) Response: The authors agree with the reviewer that our approach is simplified and it does not take into account all possible GWs at heights from the ground to 26 km. However, our objectives are to produce simplified synthetic profiles of the dominant mesoscale GW at heights of 13-20 km in agreement with mean spectral characteristics of observed mesoscale GWs, the intensity of observed GW perturbations, phase relationships as well as energy densities and horizontal phase speed. Analyses of wave parameters from simulated profiles of perturbations and comparison between observed and simulated profiles support that the simplified synthetic profiles contain the main information about the dominant mesoscale GW at heights of 13-20 km which is useful in the next section to interpret the impact of GWs on stratospheric tracers and aerosols. The discussion about the variation of Brunt Vaisala frequency with height and the horizontal wavelength has been removed. The text of section 5.3 has been revised to mention clearly our assumptions and objectives. Change: refer to P17L26-P18L13

Comment (13) The authors believe that change in wavelength mean as a function of time provides information about the time evolution of GW distribution. They agree with the reviewer's comment that a given combination of wave parameters could preferentially be observed as a function of altitude and time depending on the background wind field. Change P18L31: "A given combination of wave parameters might preferentially be observed depending on the background wind field." at the end of 5.3: P19L6 "In conclusion, the GROGRAT simulation indicates that mesoscale GWs with a whole range of parameters around the mean parameter could be excited by the front and propagate to the location of Ile du Levant at heights of 13-20 km."

Comment (14) We have modified the sentence to discuss multiple waves instead of a single wave. Change P21L22 : "The methodology is illustrated on a case study on 27 July 2013 when mesoscale inertia GWs produced by the jet-front system were identified during a jet-streak event"

Other comments: all suggested modifications have been incorporated into the revised

text (refer to the modifications highlighted in blue). For a better quality, Fig. 9a is provided.

Please also note the supplement to this comment:
http://www.atmos-chem-phys-discuss.net/acp-2015-889/acp-2015-889-AC1-supplement.pdf

———————————————————

[revised manuscript text omitted]

---

## Author Comment (AC2) · 11 May 2016

Detailed reply to comments from the two anonymous referees of manuscript "Gravity-wave effects on tracer gases and stratospheric aerosol concentrations during the 2013 ChArMEx campaign" (acp-2015-889)

Fabrice Chane Ming1, Damien Vignelles2, Fabrice Jegou2, Gwenael Berthet2, Jean-Batiste Renard2, François Gheusi3 and Yuriy Kuleshov4,5,6,7

1Université de la Réunion, Laboratoire de l'Atmosphère et des Cyclones, UMR 8105, UMR CNRS-Météo France-Université, La Réunion, France 2CNRS, LPC2E, UMR 7328, CNRS / Université d'Orléans, Orléans, France 3Laboratoire d'Aérologie,

[Figure]

UMR5560, Université de Toulouse and CNRS, Toulouse, France 4Bureau of Meteorology, Australia 5School of Science, RMIT University, Australia 6Faculty of Science, Engineering and Technology, Swinburne University, Australia 7School of Mathematics and Statistics, Melbourne University, Australia

First, we thank the anonymous reviewers for their helpful detailed comments and suggestions on our manuscript. In the following, comments of the reviewers are fully addressed and modifications have been made in the revised manuscript accordingly. Responses to comments of referee #2 are highlighted in red; those of referee #3 in blue as well as revised sentences and paragraph in the updated manuscript. A new version of the manuscript is attached as supplement.

Yours sincerely, Fabrice Chane Ming

Referee 2:

Major comments:

1.) The reviewer suggests that the GW patterns in the ECMWF analyses look like concentric. This observation is unclear and the authors think that we should be cautious about interpreting ECMWF analyses as true representation of GW patterns. However, the authors are planning to use a mesoscale model (WRF model) to assess the wave process and investigate complexity of a jet-front system as a source of GWs.

2.) This part has been modified as the other reviewer suggested. The explanation of the calculation is described in five lines and therefore it does not need to be moved to an appendix (refer to comment P5).

3.) The conclusion has been modified accordingly.

4.) The language has been improved by a native English speaker. Some additional introductory or explanatory sentences have been included in the text.

5.) We have modified the conclusion to address the reviewer's comment.

6.) The authors hope that the figure quality is now satisfactory for the reviewer. For a better quality, the Fig. 9 is provided. In addition all high quality figures can be provided separately.

6b) We have made consistency of the maps. For Fig. 6a, we can not extend the latitude range at -20°E because it is beyond the size of the image.

Specific comments: P7L11 We have specified " Background temperature and horizontal wind profiles". P7L18

P9L1 GPS-RO measurements have vertical resolution varying from about 0.5 km in the lower stratosphere (LS) to 1.4 km at 40 km heights in the middle atmosphere (refer to P6L2). So GWs with vertical wavelengths of 2-4 km can be observed. Indeed Fig. 6b shows evidence of a variability of GW energy from a vertical wavelength of 1.5 km in GPS RO profiles.

P9L28 The authors agree with the reviewer. We have reformulated the sentence. Change P10L9: "The validity of the Wentzel-Kramers-Brillouin (WKB) approximation based on the slow variation of the vertical wavenumber with height ensures the integration of the ray equations. Previous studies have shown that GROGRAT is an efficient tool to identify GW sources (Guest et al., 2000, 2002; Pramitha et al., 2015) and to simulate GW-background interactions such as GW effects, wave filtering, space and time variability of GW activity and characteristics (Wei and Zhang, 2015)."

P11L9 The modification has been done.

P11L13 The modification has been done.

P11LL29/Figure2: We have described more clearly the use of such charts. Change P12L10: "Ucellini and Koch (1987) and Koch and O'Handley (1997) produced similar charts to depict synoptic environment of jet streak typifying occurences of mesoscale GWs at the exit region of the upper-level jet streak." P12L15 " Maximum wind speeds at 300 hPa visualize the jet core at west of Portugal. In addition wind speeds >50 ms-1

from 1200 UTC indicate the presence of a significant jet streak."

Figure 4: Figures have been redrawn to address the reviewer's comments.

P14L6 We have specified what is precisely in quadrature. Hodographs of u' and v' is based on the phase quadrature between u' and v' and they have been intensively used to visualize and to extract elliptical structures. These structures are characteristic of GWs independently on the orientation of the wave vector. The orientation of the ellipse provides the direction of the wave vector ($\pm \pi$). Change P14L22: " RS horizontal wind perturbations (u' and v')"

P14LL20 and Fig 5b: In the present version, we have focused on GWs with horizontal wavelengths <1200 km observed at latitudes between 42° and 48°N (refer to new figures 5) for which the horizontal wavelengths can be correctly estimated and are clearly visible on the data series. The previous Fig 5b has been removed as redundant. Revised Figure 5 provides the main information: location of wavelike structures and presence of waves with horizontal wavelengths of 400-800 km. Indeed, some small-scale perturbations with long wavelengths could be a noise in ECMWF data.

Fig 5d: The new Fig 5c reveals a continuous spectrum of GWs with horizontal wavelengths of about 400-800 km. We can observe several peaks at latitude of 46.5° with a dominant one at about 400 km. Thus, we do not discuss subharmonics.

Change P14L1-9: " Spectral density of vertical velocities is calculated for longitudes of 10°W-20°E at latitudes between 42° and 48°N. In particular, the energy of mesoscale wavelike structures with horizontal wavelengths in the east–west direction ïĄňx<500 km is well-localized at the latitude of 46 °N which corresponds to the latitude of the exit region of the jet streak. Figure 5b displays the energy distribution of dominant wavelike structures at latitudes of 42-48°N for horizontal wavelengths ïĄňx<1200 km. In particular, it highlights a continuous spectrum of GWs with horizontal wavelengths ïĄňx of 400-800 km with a dominant mode of 400 km at latitudes of about 46°N. The spectral densities at latitudes of 42-48°N and 46°N confirm that the mode of 400 km is

dominant at 46°N within the wavelength range of 400-800 km (Fig. 5c). "

Figure 6a: The map has been redrawn to keep consistency of maps (refer to major comment 6b)

P15 The first paragraph has been revised to address the reviewer's suggestion. Change P14L12-19: "The waves from this event can be identified in GPS-RO soundings as well. Figure 6a shows an overview of the GPS-RO soundings over western Europe for the days 26 to 29 July, 2013. We selected profiles for longitudes of 2.5°-6° E and latitudes of 40-50° N for spectral analysis of the altitude range of 10-20 km. The results are shown in Figure 6b. The individual spectra are labeled by the day and UTC of the measurement (e.g. 2605 for 26 July; 05 UTC) and marked also accordingly in Figure 6a. As one can see from results for consecutive days, GWs are enhanced starting from 27 July, peak at 28 July and are still active on 29 July. Spectral density peaks are found for wavelengths 2-3.5 km"

Table 1: The methods used to obtain parameters provide characteristics of dominant GWs, e.g., peaks of wave parameter distributions. For example, at heights of 3-7 km, we have reported three values of horizontal wavelengths because the hodograph analysis provides three peaks in the $\omega$/f distribution. Change P32: "Characteristics of dominant GWs on 27 July at 2303 UTC above Ile du Levant using a hodograph analysis and combined conventional methods (grey cells) on observed and simulated (marked by *) profiles. Values are derived from peaks of wave parameter distributions."

P16LL12 The text about possible reasons for phase shift uncertainty has been removed because effects are supposed to be minor or difficult to quantify. Indeed, previously published papers on retrieval of horizontal wavelengths from GPS RO phase shift were not focused on such uncertainties and therefore in this paper we also do not discuss such uncertainties. In addition the part about phase shift has been modified according to comments of another reviewer. In the present study, time variation is taken into account because the value of $\omega$ has been previously estimated. The details of the

analysis based on five concise lines do not need to be moved into an appendix.

Change: refer to P16L28- P17L12 for modification about the computation of the phase shift in taking into account time variation. We have added an explanation, as suggested by the reviewer P17L12: "The estimate is somewhat lower but compatible with the value of 0.05 m2s-2 derived from the radiosonde measurements"

Page 18L1: The modification has been done.

P18 first paragraph: Few rays break near the tropopause in particular for large horizontal wavelengths (450-550 km). If we extend the range of $\omega$/f and horizontal wavelengths we can observe a wider dispersion in the termination altitude, time and location. The authors have verified the termination condition at the altitude of the jet core. It is not simply because of the tropopause because the tropopause is located at about 15 km heights and rays are traced back to 10 km heights. Thus, in this paper we only present the results for which we can provide explanations. In our future research, a more detailed description of the wave process will be investigated using a mesoscale model (WRF). Change P18L26: "Because rays pass over a convective region when the front system moves eastward, some GWs produced by convective sources might also be captured."

Please also note the supplement to this comment:
http://www.atmos-chem-phys-discuss.net/acp-2015-889/acp-2015-889-AC2-supplement.pdf

**Fig. 1.** Backward rays from Ile du Levant launched at 19 km height on 28 July at 0000 UTC onto (a) georeferenced infrared GMS-3 image

[Figure]

---

## Author Response (AR2)

**Reply to the editor about technical corrections before publication of the manuscript "Gravity-wave effects on tracer gases and stratospheric aerosol concentrations during the 2013 ChArMEx campaign" (acp-2015-889)**

Fabrice Chane Ming[1], Damien Vignelles[2], Fabrice Jegou[2], Gwenael Berthet[2], Jean-Baptiste Renard[2], François Gheusi[3] and Yuriy  Kuleshov[4,5,6,7]

[1]Université de la Réunion, Laboratoire de l'Atmosphère et des Cyclones, UMR 8105,  UMR CNRS-Météo France-Université, La Réunion, France

[2]CNRS, LPC2E, UMR 7328, CNRS / Université d'Orléans, Orléans, France

[3]Laboratoire d'Aérologie, UMR5560, Université de Toulouse and CNRS, Toulouse, France

[4]Bureau of Meteorology, Australia

[5]School of Science, RMIT University, Australia

[6]Faculty of Science, Engineering and Technology, Swinburne University, Australia

[7]School of Mathematics and Statistics, Melbourne University, Australia

First, the authors thank the editor and the two anonymous reviewers for their helpful detailed comments and technical suggestions on our manuscript from the submission process to the final step of the whole process and for their encouragement to improve the paper for publication in ACP journal. (refer to the acknowledgements)

All suggested modifications about technical comments from last reports (3 June 2016) of the editor and the two referees are included in the new version of the manuscript.

Yours sincerely,
Fabrice Chane Ming